# Self-organized patterning of cell morphology via mechanosensitive feedback

**Natalie A Dye**[1,2,3†*], **Marko Popović**[4,5,6†], **K Venkatesan Iyer**[1,2‡],
**Jana F Fuhrmann**[1,2], **Romina Piscitello-Gómez**[1,2], **Suzanne Eaton**[1,2§],
**Frank Jülicher**[2,5,6*]

[1]Max Planck Institute for Molecular Cell Biology and Genetics, Dresden, Germany;
[2]Cluster of Excellence Physics of Life, Technische Universität Dresden, Dresden,
Germany; [3]Mildred Scheel Nachwuchszentrum (MSNZ) P2, Medical Faculty,
Technische Universität Dresden, Dresden, Germany; [4]Institute of Physics, École
Polytechnique Fédérale de Lausanne, Lausanne, Switzerland; [5]Max Planck Institute
for the Physics of Complex Systems, Dresden, Germany; [6]Center for Systems
Biology Dresden, Dresden, Germany

**\*For correspondence:**
natalie_anne.dye@tu-dresden.de
(NAD);
julicher@pks.mpg.de (FJ)

[†]These authors contributed
equally to this work

**Present address:** [‡]Department
of Mechanical Engineering,
Indian Institute of Science,
Bangalore, India

[§]Deceased

**Competing interests:** The
authors declare that no
competing interests exist.

**Reviewing editor:** Jaume
Casademunt, University of
Barcelona, Spain

**Abstract** Tissue organization is often characterized by specific patterns of cell morphology. How such patterns emerge in developing tissues is a fundamental open question. Here, we investigate the emergence of tissue-scale patterns of cell shape and mechanical tissue stress in the *Drosophila* wing imaginal disc during larval development. Using quantitative analysis of the cellular dynamics, we reveal a pattern of radially oriented cell rearrangements that is coupled to the buildup of tangential cell elongation. Developing a laser ablation method, we map tissue stresses and extract key parameters of tissue mechanics. We present a continuum theory showing that this pattern of cell morphology and tissue stress can arise via self-organization of a mechanical feedback that couples cell polarity to active cell rearrangements. The predictions of this model are supported by knockdown of MyoVI, a component of mechanosensitive feedback. Our work reveals a mechanism for the emergence of cellular patterns in morphogenesis.

## Introduction

During morphogenesis, tissues with complex morphologies are formed from the collective interplay of many cells. This process involves spatial patterns of signaling activity, and previous work has discovered mechanisms for generating tissue-scale patterns of activity in signaling pathways such as Hedgehog, TGFβ, and Wnt (*Green and Sharpe, 2015*). In addition, patterns of cellular morphology arise during morphogenesis. Such patterns can be important for ensuring the function of the resulting tissue. For example, the compound eye of *Drosophila* consists of hundreds of ommatidia organized in a precise hexagonal array that is required to fully sample the visual field (*Kumar, 2012*). Patterns of cellular morphology that arise during morphogenesis can also guide the morphogenetic processes itself. For example, spatial patterns of cell morphology emerge during growth of the *Drosophila* larval imaginal discs, which are precursors of adult tissues (*Aegerter-Wilmsen et al., 2010*; *Condic et al., 1991*; *Legoff et al., 2013*; *Mao et al., 2013*). These patterns have been proposed to be involved in the eversion process, during which these flattened epithelial sacs turn themselves inside out when the animal transitions from larva to pupa (*Condic et al., 1991*). While extensive work has studied the emergence of biochemical signaling patterns, how patterns of cellular morphology arise during tissue development is poorly understood.

**eLife digest** During development, carefully choreographed cell movements ensure the creation of a healthy organism. To determine their identity and place across a tissue, cells can read gradients of far-reaching signaling molecules called morphogens; in addition, physical forces can play a part in helping cells acquire the right size and shape. Indeed, cells are tightly attached to their neighbors through connections linked to internal components. Structures or proteins inside the cells can pull on these junctions to generate forces that change the physical features of a cell. However, it is poorly understood how these forces create patterns of cell size and shape across a tissue.

Here, Dye, Popovic et al. combined experiments with physical models to examine how cells acquire these physical characteristics across the developing wing of fruit fly larvae. This revealed that cells pushing and pulling on one another create forces that trigger internal biochemical reorganization – for instance, force-generating structures become asymmetrical. In turn, the cells exert additional forces on their neighbors, setting up a positive feedback loop which results in cells adopting the right size and shape across the organ. As such, cells in the fly wing can spontaneously self-organize through the interplay of mechanical and biochemical signals, without the need for pre-existing morphogen gradients.

A refined understanding of how physical forces shape cells and organs would help to grasp how defects can emerge during development. This knowledge would also allow scientists to better grow tissues and organs in the laboratory, both for theoretical research and regenerative medicine.

Here, we investigate tissue-scale patterning of cell morphology in the *Drosophila* larval wing imaginal disc, which has a geometry that is ideal for studying spatial patterns of epithelial cell morphology. We focus specifically on the cell shape patterns in the central 'pouch' region, which is the precursor of the adult wing blade. To a good approximation, this region is planar and ellipsoidal. Cells near the center have smaller cell areas and are more isotropic in shape, whereas cells near the periphery have larger cell areas and are elongated tangentially (*Aegerter-Wilmsen et al., 2010*; *Legoff et al., 2013*; *Mao et al., 2013*). Cell shape has been correlated with mechanical stress: tangentially oriented bonds of elongated cells in the periphery are under higher tension than radially oriented bonds (*Legoff et al., 2013*).

It has been previously proposed that this pattern of cell morphology in the wing pouch could stem from differential proliferation: if the center grows faster than the rest, the resulting area pressure could stretch peripheral cells tangentially (*Mao et al., 2013*). Indeed, there is evidence to suggest that cells divide slightly faster closer to the center during very early stages (before $80hr$ after egg laying, AEL). It was suggested that this early growth differential is sufficient to account for the persistence of the cell morphology pattern through the remaining $\sim 40hr$ of development. However, it has since been shown that cell rearrangements occur (*Dye et al., 2017*; *Heller et al., 2016*), which could relax stress patterns once growth has become uniform. Furthermore, stress patterns may even relax during homogeneous growth in the absence of cell rearrangements (*Ranft et al., 2010*). Thus, it remains unclear how cell morphology patterns generated early by differential growth could be maintained through later stages, and alternative mechanisms for the establishment of these patterns must be considered.

Here, we measure the spatial patterns of cell morphology, cell divisions, and cell rearrangements during the middle of the third larval instar (starting at $96hr$ AEL). We quantify the pattern of tangential cell elongation and show that it becomes stronger over time, even though growth is spatially uniform and cell rearrangements are frequent. Strikingly, this change in tangential cell elongation is coupled to a radially biased pattern of cell neighbor exchanges. Using a physical model of tissue dynamics, we show that active patterning of radial cell neighbor exchanges can account for the observed morphology patterns in the absence of differential growth. Lastly, using a combination of experiment and theory, we provide evidence that this active patterning could be self-organized by mechanosensitive feedback.

## Results

### Cell morphology patterns can persist and strengthen in the absence of differential growth

Cell morphology patterns in the wing disc have been previously analyzed using static images (*Aegerter-Wilmsen et al., 2010*; *Legoff et al., 2013*; *Mao et al., 2013*). However, relating cell morphology patterns to patterns of growth, cell divisions, and cell rearrangements requires dynamic data. We therefore performed long-term timelapse imaging of growing explanted wing discs using our previously described methods (*Dye et al., 2017*), starting at $96hr$ AEL and continuing for $\sim 13hr$ of imaging. We used Ecadherin-GFP as an apical junction marker (*Figure 1A*).

To quantify cell morphology, we averaged apical cell area and cell elongation locally in space, using data from five wing discs, and in time using $\sim 2hr$ intervals. Over the course of the $13hr$ timelapse, the qualitative features of the morphological patterns do not change. Therefore, in *Figure 1B–C*, we present the spatial patterns calculated for the middle timepoint. Cell area is represented as a color code. Cell elongation is characterized by a tensor $\boldsymbol{Q}$, which defines an axis and a strength of elongation and is represented by bars in *Figure 1C* (see Materials and methods, *Figure 1—figure supplement 1B*; *Etournay et al., 2015*; *Merkel et al., 2017*). To quantify the radial symmetry of this pattern, we first determined the center of symmetry (*Figure 1—figure supplement 1* and Materials and methods). We then introduce a polar coordinate system at the center with the radial coordinate $r$ and present the radial projection $Q_{rr}$ of the cell elongation tensor as a color code in *Figure 1C* (see also *Figure 1—figure supplement 1*). This figure highlights the pattern of tangential cell elongation, with cells elongating on average perpendicular to the radial axis (blue in *Figure 1C*). It also reveals that this pattern is interrupted around the Dorsal-Ventral (DV) boundary, where cells are elongated parallel to this boundary (red in *Figure 1C*). We quantify this region separately (*Figure 1—figure supplement 2*) and exclude it from our analysis of the circular patterns (*Figure 1D–F*). The Anterior-Posterior (AP) boundary also affects cell morphology (*Landsberg et al., 2009*), but the effect is weaker and more variable at this stage; thus, we do not quantify it separately.

The spatial maps of cell morphology reveal that both cell area and cell elongation magnitude are largest at the periphery and decrease toward the center (*Figure 1B–C*). We quantified this radial gradient in cell area and observe that cell area ranges from $\sim 3-7\mu m^2$ when moving toward the periphery (*Figure 1E*). We also observe a radial gradient in cell elongation starting from $Q_{rr} \approx 0$ at $r = 10\ \mu m$ and extending to about $Q_{rr} \approx -0.1$ at $r = 40\ \mu m$ (*Figure 1F*). The negative value corresponds to tangential elongation (*Figure 1C*). When evaluated over the timelapse, we find that these radial gradients grow slightly more pronounced over time (*Figure 1—figure supplement 3C–D*).

As previously proposed, differential growth can generate such patterns of cell elongation (*Mao et al., 2013*). However, indirect metrics of tissue growth *in vivo* do not indicate that differential growth still occurs at this later stage (*Mao et al., 2013*). We directly measured the spatial pattern of growth during timelapse. Cell division rate has been previously used as an indicator of growth; however, tissue growth actually results from a combination of cell division, cell area changes, and cell extrusions (*Figure 1G*). Thus, we evaluated the spatial pattern (*Figure 1H,I,K,L*) and radial profiles (*Figure 1J,M*) of total tissue growth and its cellular contributions. These data show that tissue area growth, as well as cell division rate, are to a good approximation independent of the distance to the center.

In summary, we quantified cell morphology patterns in mid-third instar wing explants during live imaging. We confirm previous static observations of the pattern and further identify a region around the DV boundary with a morphological pattern that differs from the rest of the wing pouch. We quantify the radial gradients in cell area and cell elongation existing outside of the DV boundary region and show that they strengthen in time in the absence of differential growth, raising the question of what mechanism underlies the persistence of these cell morphology patterns during mid to late stages of wing growth.

### Radially oriented cell rearrangements balance tangential cell elongation

To directly relate the observed cell morphology patterns with cell rearrangements, we next analyzed the spatial patterns in cellular contributions to tissue shear (*Figure 2A*). Tissue shear can be

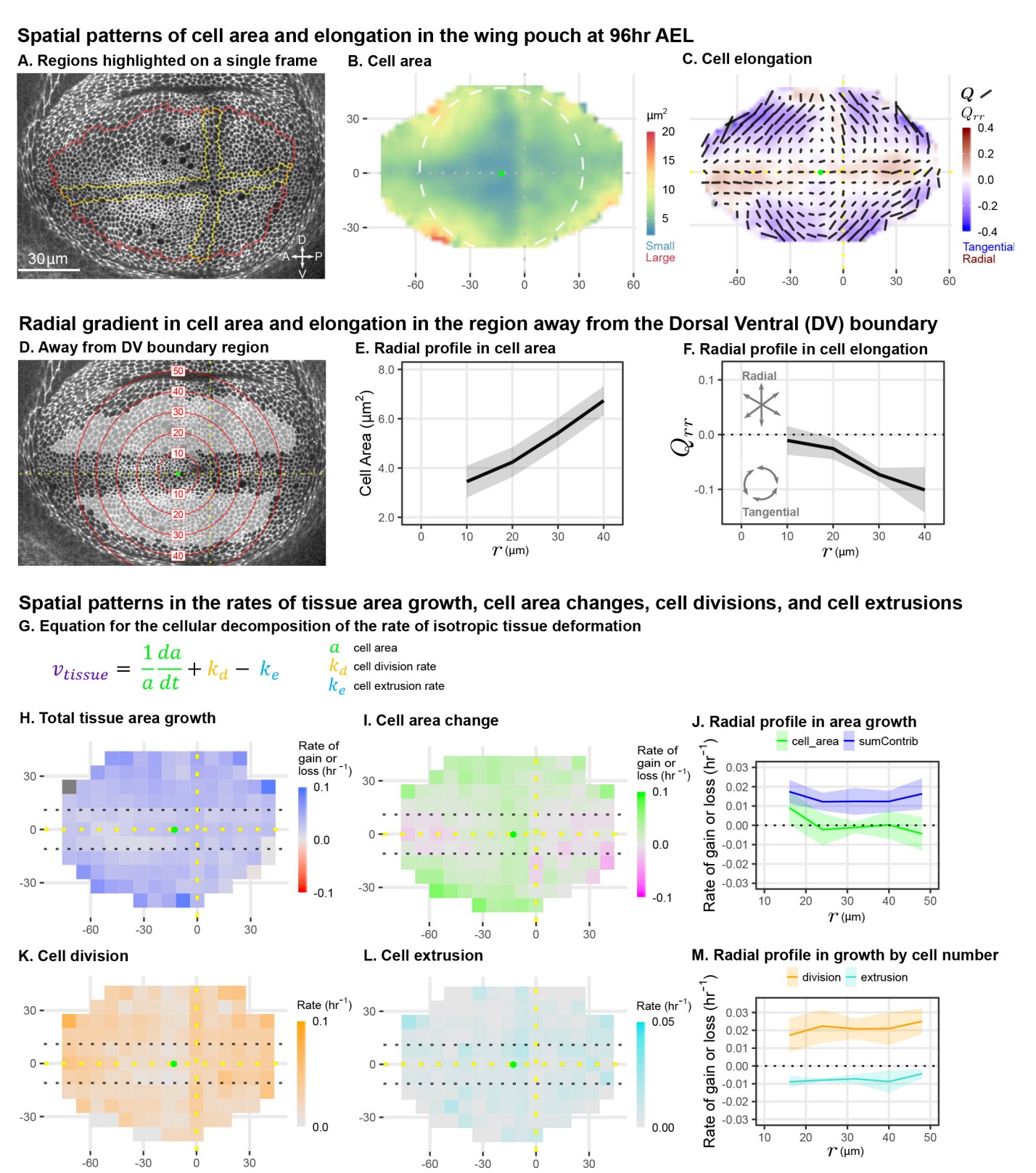

**Figure 1.** Cell morphology patterns can persist and strengthen in the absence of differential growth. (**A**) Ecadherin-GFP-expressing wing disc growing in *ex vivo* culture (explanted at *96hr* AEL). We analyze apical cell morphology in the proliferating disc proper layer in a 2D projection, after correcting

*Figure 1 continued on next page*

*Figure 1 continued*

for local tissue curvature (see Materials and methods and *Figure 1—figure supplement 1A*). Red outline indicates the presumptive 'blade' region; yellow outline indicates cells of the compartment boundaries, which are used to align different movies. Dorsal is up, and anterior is left. (B–C) Spatial maps of cell area (B) or cell elongation (C) were generated by averaging across the middle timepoints of five movies. Axis labels indicate the distance in $\mu m$ from the AP boundary (along X) or DV boundary (along Y). Dotted lines indicate the compartment boundaries. The calculated center of symmetry is represented by a green dot. In (B), the white dashed circle is added to highlight the near-circular symmetry. In (C), the bars represent the cell elongation tensor $Q$, where the length of the bar is proportional to the magnitude and the angle indicates its orientation. In addition, the radial component of cell elongation is presented in color. More information about this cell elongation pattern and how we define the center point is included in *Figure 1—figure supplement 1*. (D) The blade cells that are tangentially elongated and lying well outside the DV boundary region are shaded in gray on a single image of the timelapse. Red circles indicate the radial binning used in (E–F), where the numbers indicate the radius (in $\mu m$). The region around the DV boundary was separately analyzed in *Figure 1—figure supplement 2*. For the region depicted in (D), the radial gradient in average cell area (E) and radial cell elongation $Q_{rr}$ (F) were calculated. Solid lines indicate the average over all five movies in the middle time window. The shaded region indicates the standard deviation. Data showing how these radial gradients change over time during imaging are shown in *Figure 1—figure supplement 3*. Source data for (B–F) are available on Dryad. (G) Decomposition of isotropic tissue deformation into cellular contributions. (H–I, K–L) Maps of the total tissue area growth (H) and its contributions from cell area change (I), cell divisions (K), and cell extrusions (L) do not show pronounced spatial patterns. (J, M) The radial profile of tissue growth and its cellular contributions, analyzed in radial bins. sumContrib in blue is the sum of all contributions, corresponding to total tissue deformation. Source data files for isotropic tissue deformation are provided as *Figure 1—source data 1*, *2*.
The online version of this article includes the following source data and figure supplement(s) for figure 1:

**Source data 1.** Isotropic contributions to tissue deformation calculated in a grid.
**Source data 2.** Isotropic contributions to tissue deformation calculated radially.
**Figure supplement 1.** Quantification of the radial elongation pattern.
**Figure supplement 2.** Cell elongation and dynamics around the DV boundary.
**Figure supplement 3.** Change in cell area and elongation during timelapse.

decomposed into contributions from cell divisions, cell elongation changes, T1 transitions, and so-called correlation effects (*Etournay et al., 2015*; *Merkel et al., 2017*). Here, correlation effects result mainly from correlated fluctuations in cell elongation and cell rotation (see Appendix 3 and *Merkel et al., 2017*).

We find that the spatial patterns of tissue growth and its cellular contributions exhibit overall anisotropies perpendicular to the DV boundary, as reported previously (*Dye et al., 2017*). In addition, the patterns of cell elongation changes and T1 transitions can be described as a superposition of a uniformly oriented pattern and an approximately radial or tangential pattern (*Figure 2C–D*). To determine the magnitude of the radial or tangential patterns in all quantities, we quantified their average radial projections as cumulative plots over time (*Figure 2B*). The radial component of tissue shear is small, and cell divisions do not contribute to radial tissue shear. In contrast, we observe a pronounced buildup of a tangential pattern of cell elongation accompanied by a radial pattern of T1 transitions and of correlation effects.

As shown above (*Figure 1J*), tissue area growth does not have a radial gradient and thus does not contribute to the increase in tangential cell elongation that we observe at this time (*Figure 2A–B*, *Figure 1—figure supplement 3C–D*). Furthermore, we observe numerous T1 transitions (on average $1.0\ cell^{-1}\ hr^{-1}$), and their radially biased orientation increases rather than relaxes tangential cell elongation (*Figure 2E–F*). Thus, we are not observing the relaxation of a pattern of cell elongation caused by early differential growth. Rather, our data support a model whereby a radially patterned morphogenetic cue actively biases the direction of T1 transitions and consequently the complementary pattern of cell shape changes.

## Polarity-driven cell rearrangements can create the observed cell morphology patterns in the wing disc

We next apply a biophysical model to determine whether radially patterned T1 transitions could account for the observed cell morphology patterns in the wing disc. This model takes into account the interplay of T1 transitions, cell shape changes, and tissue shear in a continuum description (*Etournay et al., 2015*; *Popović et al., 2017*). Active anisotropic force-generating processes that bias cell rearrangements in the tissue, such as polarization of the actomyosin cytoskeleton, are captured by a nematic cell polarity $q$, defined by a magnitude and an orientation axis. We propose that such a patterning cue leads to the radially oriented pattern of T1 transitions we observe in the wing disc.

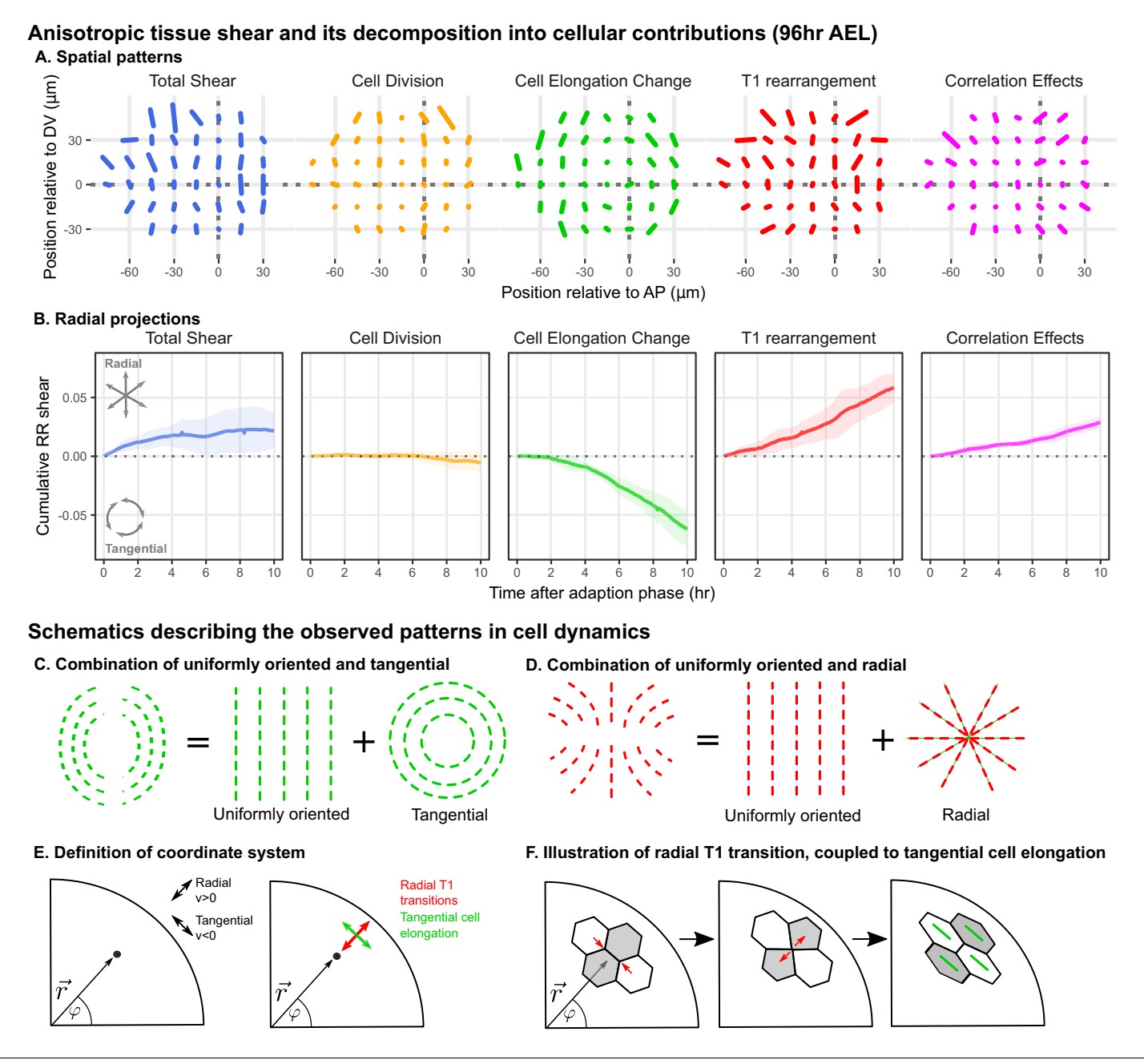

**Figure 2.** Radially oriented cell rearrangements balance tangential cell elongation. (**A**) Cumulative tissue deformation and its cellular contributions, measured in a grid centered on the compartment boundaries (dotted gray lines) and averaged over all five movies. Bars represent nematic tensors, where the length is proportional to the magnitude of deformation and the angle indicates its orientation. The contribution from cell extrusion is small and thus not shown. Data used to present these plots are included in *Figure 2—source data 1*. (**B**) The radial projection of cumulative tissue deformation and its cellular contributions are plotted as a function of time after the first 2*hr* of adaption to culture (*Dye et al., 2017*). Solid lines indicate the average over all five movies; shading indicates the standard deviation. Data used to present these plots are included in *Figure 2—source data 2*. (**C**) Schematics indicating how a uniformly oriented pattern would combine with a tangential pattern to produce a pattern resembling that of the cell elongation change (left) or with a radial pattern to produce a pattern resembling that of the T1 transitions (right). (**E–F**) Illustrations demonstrating radially-oriented T1 transitions coupled to tangential cell elongation. For simplicity, we diagram only the posterior-dorsal quadrant, but the pattern is radially symmetric. In (**E**), we define the radial coordinate system, where velocity in the radial direction is positive and that in the tangential direction is negative. Patterns in A indicate that T1 transitions are biased to grow new bonds in the radial direction, and cell elongation is biased to increase tangentially. (**F**) In a radially oriented T1 transition, cells preferentially shrink tangentially oriented bonds and grow new bonds in the radial direction. When oriented in this direction, T1 transitions do not dissipate tangential elongation but increase it (green bars in each cell represent cell elongation).

*Figure 2 continued on next page*

*Figure 2 continued*

The online version of this article includes the following source data for figure 2:

**Source data 1.** Anisotropic contributions to tissue deformation calculated in a grid.

**Source data 2.** Cumulative radial shear plots over time.

In our model, we consider the spatial patterns of tissue shear rate $\tilde{v}$, the patterns of cell shape $\boldsymbol{Q}$, and the patterns of cell rearrangements $\boldsymbol{R}$, which have nematic symmetry and are represented by traceless symmetric tensors that describe the magnitude and orientation. Tissue shear is defined as a velocity gradient tensor that results from a combination of cell shape changes and cell rearrangements (*Etournay et al., 2015*; *Merkel et al., 2017*):

$$\tilde{v} = \frac{\mathrm{D}\boldsymbol{Q}}{\mathrm{Dt}} + \boldsymbol{R} \tag{1}$$

Here, $\mathrm{D}\boldsymbol{Q}/\mathrm{Dt}$ is a co-rotational time derivative of $\boldsymbol{Q}$, and the shear due to cell rearrangements $\boldsymbol{R}$ includes contributions from T1 transitions, cell divisions and extrusions, and also correlation effects.

Tissue material properties are described by constitutive equations for the tissue shear stress $\tilde{\sigma}$ and the shear due to cell rearrangements $\boldsymbol{R}$ (*Figure 3A–B*):

$$\tilde{\sigma} = 2K\boldsymbol{Q} + \zeta\boldsymbol{q} \tag{2}$$

$$\boldsymbol{R} = \frac{1}{\tau}\boldsymbol{Q} + \lambda\boldsymbol{q} \tag{3}$$

Here, the shear stress tensor $\tilde{\sigma}$ is the sum of the elastic and active stress. The elastic stress is associated with cell elongation $\boldsymbol{Q}$ and characterized by the shear elastic modulus $K$. The active stress associated with the cell polarity cue $\boldsymbol{q}$ (*Figure 3A*) is described by the coefficient $\zeta$. The shear rate due to cell rearrangements $\boldsymbol{R}$ given by *Equation 3* is driven in part by shear stress, and therefore depends on the cell elongation $\boldsymbol{Q}$, and in part by active processes that are oriented by the nematic cell polarity $\boldsymbol{q}$ (*Figure 3B*). $\tau$ is a relaxation time for cell rearrangements over which elastic stresses are relaxed, and $\lambda$ is the rate of cell rearrangements driven by cell polarity. In this coarse-grained picture, subcellular processes are captured by effective coefficients. For simplicity, we do not include dissipative processes on cellular scales, such as cytoskeletal viscosity or cell-cell friction. Such dissipative processes are relevant on short timescales and are small in comparison with elastic stresses on tissue relevant timescales.

To discuss the wing disc, we consider a radially symmetric geometry and average the oriented quantities after projection onto the radial axis. Radial tissue shear is small compared to that associated with cell shape changes and T1 transitions during our observed time window (*Figure 2B*). We therefore consider a steady state with $\tilde{v}_{rr} = 0$, $\mathrm{D}Q_{rr}/\mathrm{Dt} = 0$, and $R_{rr} = 0$. In this case, cell elongation becomes:

$$Q_{rr} = -\tau\lambda q_{rr} \tag{4}$$

Thus, we find that the steady state cell elongation pattern is a result of cell rearrangements that are oriented by the cell polarity cue $\boldsymbol{q}$ (*Figure 3C–D*). Note that our data show that the wing disc is not exactly at steady state: cells slowly change their shape and rearrange radially (*Figure 2A–B*). However, as we show in Appendix 1 part 1-2, *Equation 4* holds to a good approximation.

Can the radial pattern of T1 transitions defined by $\boldsymbol{q}$ also explain the observed radial profile of cell area (*Figure 1B,E*)? To answer this question, we then considered force balances in the tissue. We consider tissue area pressure:

$$P = -\overline{K}\ln\left(\frac{a}{a_0}\right) \tag{5}$$

where $P$ is the difference in pressure from a reference value, $\overline{K}$ is tissue area compressibility, $a$ is the average cell area, and $a_0$ is a reference cell area. As pressure increases, cell area decreases. To

## Continuum equations for epithelial tissue dynamics

### A. Tissue stress is composed of elastic and active stress

$$\tilde{\sigma} = 2K\mathbf{Q} + \zeta\mathbf{q}$$

Cell shear stress Cell elongation Nematic cell polarity

Elastic stress Polarity-driven

### B. Cell rearrangements are guided by cell shape and cell polarity

$$\mathbf{R} = \frac{1}{\tau}\mathbf{Q} + \lambda\mathbf{q}$$

Rearrangements Cell elongation Nematic cell polarity

Relaxation Polarity-driven

## Radially oriented cell rearrangements can generate a radial profile in cell elongation and area

### C. Radial polarity field

$$\lambda q_{rr} > 0$$

Cell Nematic Polarity

### D. Tangential cell elongation

$$Q_{rr} = -\tau\lambda q_{rr}$$

Cell elongation

### E. Pressure profile

Stress $\quad \tilde{\sigma}_{rr} = -(2K\tau\lambda - \zeta)q_{rr}$

Pressure $\quad \partial_r P = \partial_r \tilde{\sigma}_{rr} + \frac{2}{r}\tilde{\sigma}_{rr}$

### F. Cell elongation profile used to estimate q

$Q_{rr}$ vs $r$ (μm)

- - - empirical fit
— experiment

### G. Cell area profile generated by q

Cell area μm$^2$ vs $r$ (μm)

— model
— experiment

**Figure 3.** Polarity-driven cell rearrangements can create the observed cell morphology patterns. (A) Stress is a combination of elastic and polarity-driven stresses. Similarly, in (B), cell rearrangements occur to relax a stretched cell shape or to respond to an internal nematic cell polarity cue. In the cartoons in (A)-(B), we chose to depict scenarios where $\zeta$ and $\lambda$ are >0. (C–G) We apply this model to a radially symmetric tissue at steady state to approximate the wing disc. If we impose a radial polarity field (C), cell elongation is oriented in the opposite direction, according to the equation in (D) when $R_{rr} = 0$ (steady state). (E) Considering force balance in the tissue, our model also predicts a pressure profile with higher pressure in the center. (F) To fit experimental data in the wing disc, we estimate the radial profile of $q_{rr}$ from *Equation 4* by measuring cell elongation as a function of $r$ in the last ($\sim 5hr$) of the timelapse. We solve for $q_{rr}$ by making an empirical fit to this cell elongation data (see Appendix 1 part 2). (G) The cell area distribution we observe in the wing disc is consistent with the pressure profile predicted by our model (E and Appendix 1 part 2).

calculate the cell area profile, we again approximate the wing pouch as a radially symmetric disc. In the radially symmetric geometry, force balance can be expressed as:

$$\partial_r P = \partial_r \tilde{\sigma}_{rr} + \frac{2}{r}\tilde{\sigma}_{rr} \tag{6}$$

A radial profile of pressure determined from this equation implies a radial pattern of cell area via *Equation 5* (see *Figure 3C–E* and Appendix 1 part 1-2). To test this implication, we first quantify the

radial profile of cell elongation $Q_{rr}$ to estimate the profile of $q_{rr}$ using *Equation 4* (*Figure 3F*). We represent the cell elongation data in a functional form using an empirical power law that is fit to the data. Then, using this functional form in *Equations 2, 5, and 6*, we solve for the cell area profile (*Figure 3G* and Appendix 1 part 1-2). Finally, we show that this function can account for the observed pattern of cell area (see *Figure 3G*).

From this analysis, we conclude that the cell morphology patterns observed in the wing disc could be generated by radially biased cell rearrangements. Next, we test whether the stress profile predicted by the model (*Equation 2*) exists in the tissue, and we measure key mechanical parameters of the model. Later, we address the potential molecular origin of the cell polarity cue orienting the cell rearrangements.

## Circular laser ablation reveals patterns of tissue stress

Our model predicts a stress pattern in the wing disc that results from active processes that are radially oriented by a cell polarity cue. To compare this prediction to experiment, we infer tissue stress using laser ablation. Tissue stress has been estimated previously by laser ablation techniques that are based on determining the initial retraction velocity (*Bonnet et al., 2012*; *Etournay et al., 2015*; *Farhadifar et al., 2007*; *Legoff et al., 2013*; *Mao et al., 2013*; *Shivakumar and Lenne, 2016*). However, to compare theory and experiment, ideally one should measure quantities that are well-captured by the model. Therefore, instead of using initial retraction velocity, we perform circular cuts and analyze the final, relaxed position of the inner and outer elliptical contours of tissue formed by the cut (*Figure 4A*, Appendix 2). From the size and the anisotropy of the cut, we can infer anisotropic and isotropic tissues stress, normalized by the respective elastic constants, as well as the ratio of elastic constants (see Appendix 2). Furthermore, we can also infer the existence of polarity-driven stress. We name this method ESCA (*Elliptical Shape after Circular Ablation*).

We perform local measurements of tissue mechanics by cutting the tissue in the smallest possible circle that would still allow us to measure the shape of the inner piece left by the cut (radius = $7 \mu m$, encircling $\sim 5 - 15$ cells, *Figure 4A*). To relate measurements of tissue stress to cell elongation, we calculate the average cell elongation in the ablated region before it is cut (*Figure 4A*). In *Figure 4B*, we present the measured cell elongation and shear stress tensors at the position of ablation. We find that local average cell elongation correlates well with the principal direction of shear stress. Also, we observe that the cells in the band around the DV boundary, which are exposed to high Wg and Notch signaling, have different mechanical properties than elsewhere in the tissue. Near the DV boundary, cells elongate less than outside this region for comparable amounts of stress (*Figure 4B* and *Figure 4—figure supplement 1B,E*). The ratio of elastic constants in this region is also smaller: near the DV boundary $2K/\overline{K} = 2.3 \pm 0.3$, whereas outside this region, $2K/\overline{K} = 3.4 \pm 0.4$ (see also *Figure 4—figure supplement 1C,F*). We focus hereafter on the radial patterns of elongation and stress outside of the DV boundary region.

The relationship between cell elongation and stress normalized by the elastic modulus has a slope 1 in the absence of polarity-driven stress (see *Equation 2*). We observe a much smaller slope for this relationship in our data (*Figure 4C*), indicating that polarity-driven stress is significant. We now use these data to estimate the parameters of our mechanical model. We write the shear stress defined in *Equation 2* in terms of cell elongation and cell rearrangements, eliminating the orientational cue $q_{rr}$ using *Equation 3*. For the radial components, we have:

$$\tilde{\sigma}_{rr} = 2K^* Q_{rr} + 2(K - K^*)\tau R_{rr} \qquad (7)$$

where $K^* = (1 - \zeta/(2K\tau\lambda))K$ is an effective shear elastic coefficient. The difference between $K^*$ and $K$ depends on the parameters $\zeta$ and $\lambda$ associated with the nematic cell polarity. We fit *Equation 7* to the data and find $K^*/K = 0.05 \pm 0.02$ and $(1 - K^*/K)\tau R_{rr} = 0.011 \pm 0.002$ (*Figure 4C*, see Appendix 1 part 3). Combined with data from *Figure 1*, we find an estimate for the tissue relaxation time $\tau = 2 \pm 2 \, hr$, which is roughly consistent with that found during pupal morphogenesis (*Etournay et al., 2015*). From our data, we can also infer the radial profile of tissue area pressure, revealing that pressure increases toward the center (*Figure 4D* and Appendix 1 part 3). This finding is consistent with the observed cell area profile, with smaller cell areas toward the center (*Figure 1B, E*).

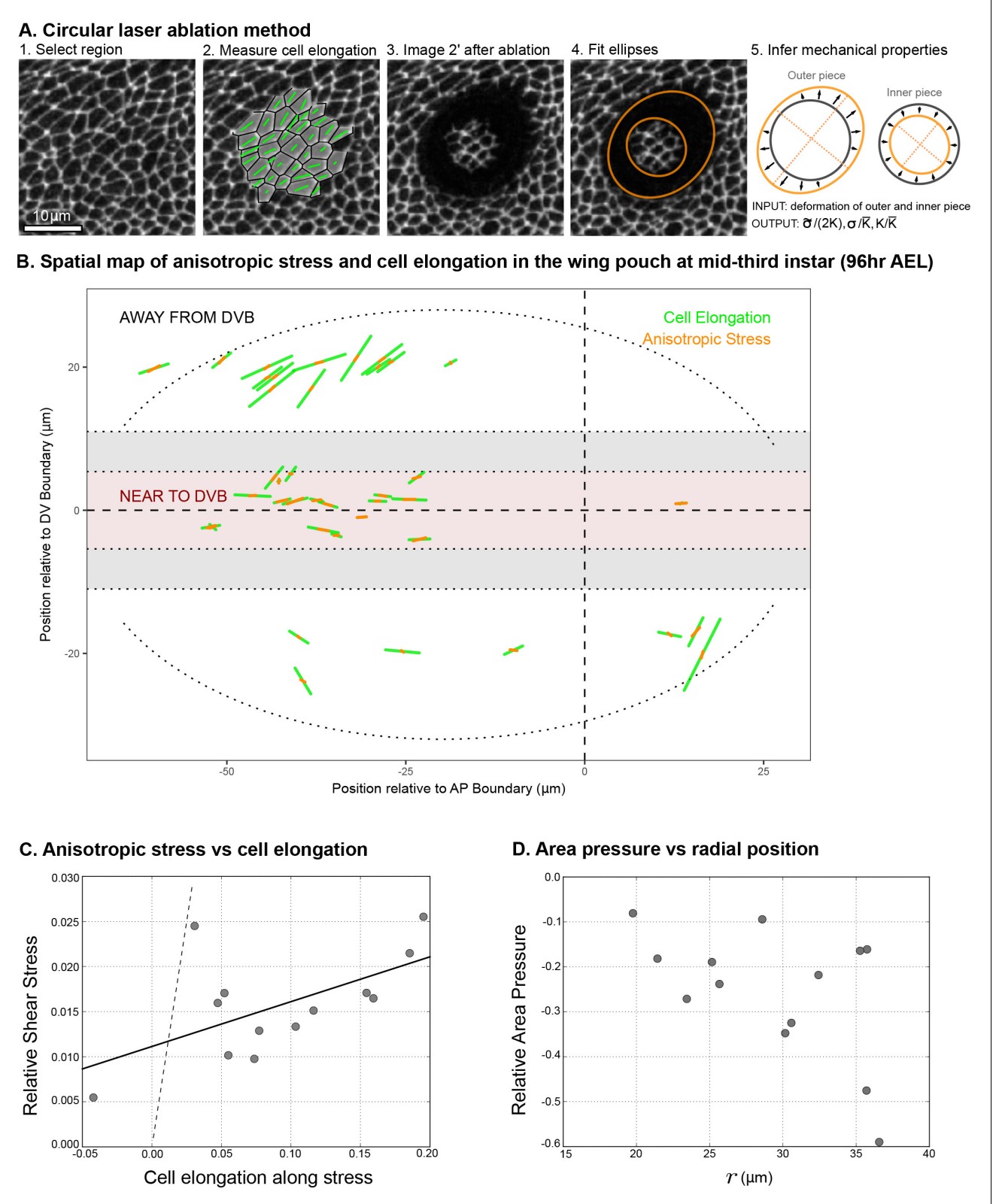

**Figure 4.** Circular laser ablation reveals patterns of tissue stress. (**A**) Description of the circular laser ablation method ESCA. The tissue is cut in a circle (radius 7$\mu m$), and cell elongation is averaged in this region before the cut. After the tissue relaxes (2$min$), we fit ellipses to the cut region. We then infer mechanical properties using our model, which inputs deformation and outputs stress as a function of elastic constant and the ratio of the isotropic and anisotropic elastic constants (see also **Figure 4—figure supplement 1**). (**B**) Map of stress and cell elongation in the wing disc (96$hr$ AEL). Each line

*Figure 4 continued on next page*

*Figure 4 continued*
represents a nematic, where length indicates the magnitude, and angle indicates its orientation. Each data point comes from a different wing disc. The gray region indicates a border region, in which a cut would straddle the two regions. Cuts centered in this region were not included in the analysis. The dotted lines on either side of the DV boundary indicate our cut-offs for delineating the border and DV boundary regions (see Materials and methods). (C) Magnitude of anisotropic stress ($\bar{\sigma}/2K$) is plotted against cell elongation (projected onto the stress axis) for all ablations outside the DV boundary. Dotted line indicates a line with slope = 1, corresponding to a tissue lacking a nematic cell polarity cue. Fit line is solid black (See Appendix 2). Data for the DV boundary region is presented in *Figure 4—figure supplement 1D-F*. (D) Relative area pressure is plotted against $r$ for all ablations outside the DV boundary. The correlation coefficient = −0.52. Data used in this figure are included in *Figure 4—source data 1*.
The online version of this article includes the following source data and figure supplement(s) for figure 4:

**Source data 1.** Laser ablation data on cell elongation and stress.
**Figure supplement 1.** Analysis of circular laser ablations (96*hr* AEL).

In sum, we find a stress profile in the wing disc that is consistent with the observed measurements of both cell elongation and area. Further, we use these data to measure certain parameters of our biophysical model, including the tissue relaxation timescale and the effective shear elastic coefficient.

## Reduction of planar cell polarity pathways does not reduce tangential cell elongation

In our model, the radial orientation cue is required to generate the observed patterns of cell morphology, cell rearrangement, and tissue stress. Candidates for such an orientational cue are the planar cell polarity pathways (PCP), which are groups of interacting proteins that polarize within the plane of the epithelium. There are two well-characterized PCP pathways: Fat and Core (*Butler and Wallingford, 2017*; *Eaton, 2003*). In the wing, these systems form tissue-scale polarity patterns during growth (*Brittle et al., 2012*; *Merkel et al., 2014*; *Sagner et al., 2012*) and are required to position the hairs and cuticle ridges on the adult wing (*Adler et al., 1998*; *Doyle et al., 2008*; *Eaton, 2003*; *Gubb and García-Bellido, 1982*; *Hogan et al., 2011*). To determine whether either of these pathways could function as the orientational cue described in our model, we analyzed cell elongation patterns after their removal.

We perturbed the Fat pathway using *nub-Gal4* to drive the expression of RNAi constructs targeting both fat (*ft*) and dachs (*d*) in the pouch region throughout the third larval instar. This perturbation results in almost complete loss of Dachsous from the apical membrane, and any residual signal is no longer polarized, confirming the loss of PCP (*Figure 5—figure supplement 1A–B*). Furthermore, we observe a suppression of tissue growth upon *ft+d* double RNAi knockdown (visible at the end of larval development in *Figure 5A* and in the resulting adult wings in *Figure 5—figure supplement 1C*), consistent with previous work on the loss of both Dachs and Fat (*Cho and Irvine, 2004*). Nonetheless, the pattern of tangential cell elongation persists to the end of larval development (*Figure 5A–C*). Using scaled coordinates, we find that the radial profiles of cell elongation in *ft+d* RNAi and control wings are similar (*Figure 5D*).

We perturbed the Core PCP pathway using a previously characterized null mutation in prickle (*pk³⁰*), which causes defects in adult wing hair orientation (*Gubb et al., 1999*). We found that the cell elongation pattern in the *pk* mutant is similar to the wild-type control (*Figure 5E–I*). In the *pk* mutant, the region of tangential cell elongation extends even further into the center than in control wings.

We conclude that the tangential cell elongation pattern persists in the absence of either PCP pathway. This result excludes these pathways as orienting cues for the cell elongation patterns.

## Mechanosensitive feedback generates self-organized patterns of cell morphology

We have shown that perturbing PCP pathways does not affect the radial patterns of morphology, raising the question of how orientational cues might arise. In previous sections, we have considered the orientational cue to be provided by a cell polarity system that is independently patterned. However, cell polarity in general would be affected by stresses in the tissue. Indeed, there are many examples of cells polarizing in response to mechanical stress (*Duda et al., 2019*; *Hirashima and Adachi, 2019*; *Ladoux et al., 2016*; *Ohashi et al., 2017*). Here, we show that introducing

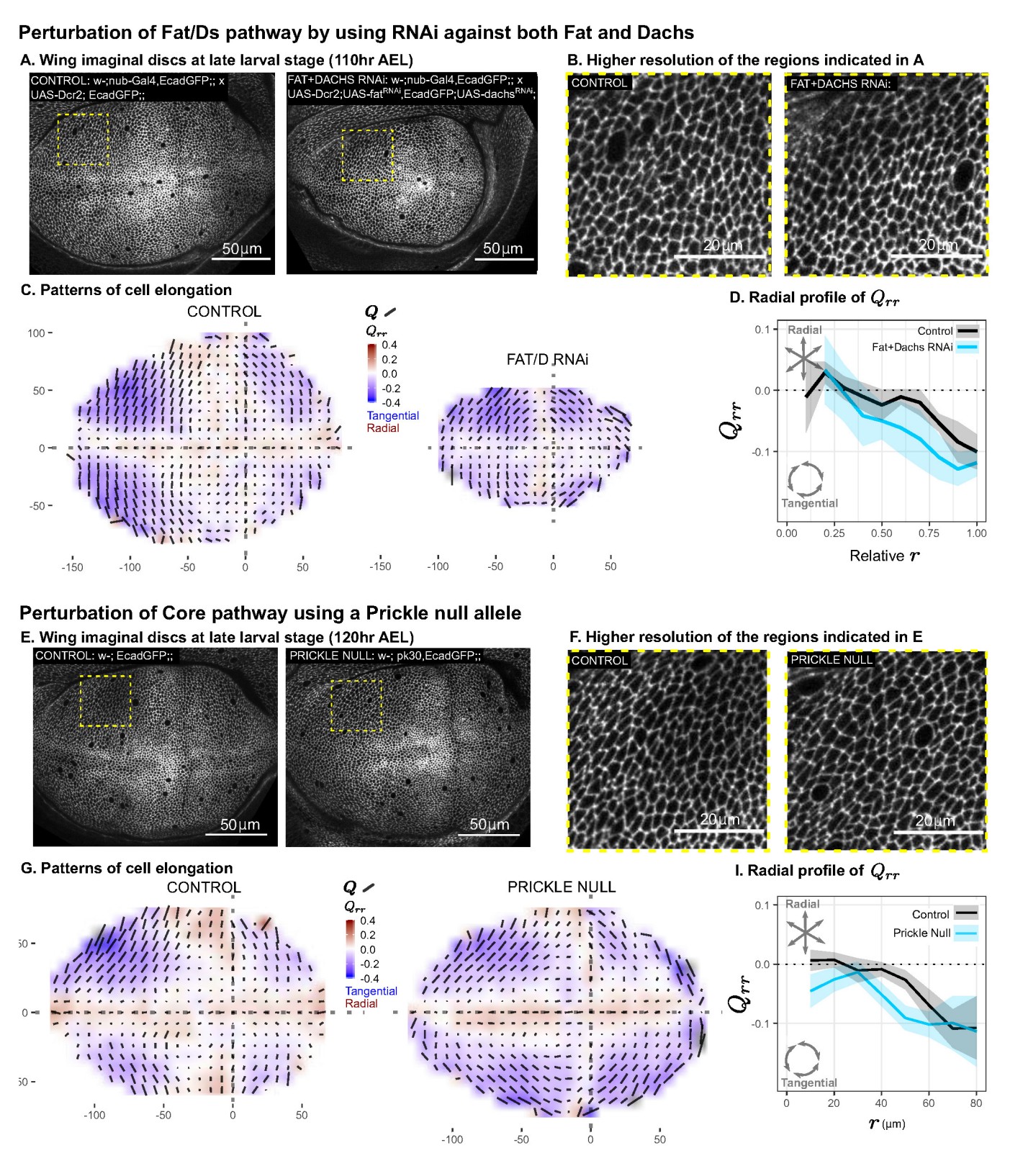

**Figure 5.** Disruption of two PCP pathways does not reduce tangential cell elongation. (A–D) RNAi was induced against both fat and dachs in the pouch (*nub-Gal4 > UAS-Dcr*, *UAS-fat*[RNAi], *UAS-dachs*[RNAi]), and cell elongation patterns at the end of larval development (110*hr* AEL) are presented along with control (*nub-Gal4 > UAS-Dcr*). Data are included in *Figure 5—source data 1*. (E–I) Cell elongation patterns at the end of larval development are

*Figure 5 continued on next page*

*Figure 5 continued*

presented for a null mutant in prickle (*pk³⁰*) and the wild-type control. Data are included in *Figure 5—source data 2*. (A,E) Representative images of each genotype, with apical cell boundaries marked with Ecadherin-GFP. Yellow box indicates the inset that is presented in higher resolution in (B,F). (C, G) Cell elongation averaged across several discs for each genotype in a grid centered on the AP and DV boundaries. Bars represent the average cell elongation tensor *Q*, where the length of the bar is proportional to the magnitude of cell elongation, and the angle indicates its orientation. Color indicates the radial component of cell elongation $Q_{rr}$. Axis labels indicate distance from the AP (X) or DV (Y) compartment boundaries (in $\mu m$). (D,I) The radial profile of cell elongation for each genotype was quantified by averaging $Q_{rr}$ in radial bins and plotting as a function of *r*. Since the *ft+d* RNAi wings are smaller, we present this profile as a relative distance to the center. The band of cells around the DV boundary was removed, and because the *ft+d* RNAi discs are smaller, this region of exclusion is a larger relative distance. Plots in (C–D) represent averages of $N = 7 - 8$ wing discs per genotype; for plots in (G–I), $N = 11 - 14$ per genotype.

The online version of this article includes the following source data and figure supplement(s) for figure 5:

**Source data 1.** Triangle elongation for Fat+Dachs RNAi.
**Source data 2.** Triangle elongation for Pk30.
**Source data 3.** Calculated center of elongation symmetry for genetic perturbations.
**Figure supplement 1.** Confirmation of reduced PCP upon *ft+d* RNAi.

mechanosensitive feedback to the model of tissue mechanics can give rise to spontaneous emergence of the cell polarity cue (*Figure 6*).

Mechanosensitivity is incorporated into our model of tissue mechanics through a dynamic equation for the orientational cue *q* that becomes stress-dependent:

$$\frac{d\boldsymbol{q}}{dt} = -\frac{1}{\tau_q}\boldsymbol{q} - \mu\tilde{\sigma} - \alpha|\boldsymbol{q}|^2\boldsymbol{q} + D\nabla^2\boldsymbol{q} \tag{8}$$

Here, $\tau_q$ is a relaxation timescale for *q*, $\mu$ is a mechanosensitive feedback strength, the coefficient $\alpha > 0$ ensures stability, and *D* is a coupling strength locally aligning orientational cues.

Now, *Equation 8* provides a mechanosensitive feedback to *Equations 1, 2 and 3*. These combined equations show a novel behavior. Specifically, the orientational cue can emerge spontaneously by self-organization (*Figure 6A–B*). Beyond a critical value $\mu_c$ of the mechanosensitive feedback strength $\mu$, an isotropic tissue with $\boldsymbol{q} = 0$ is no longer stable, and a state with an orientational cue $\boldsymbol{q} \neq 0$ emerges instead (*Figure 6B* and Appendix 1 part 4). The magnitude of this spontaneous polarization is $|\boldsymbol{q}| = q_0$, where $q_0^2 = (\tau_q\mu(2K\tau\lambda - \zeta) - 1)/(\alpha\tau_q)$, where a positive coefficient $\alpha$ is needed to stabilize the polarized state. By this mechanism, the anisotropic cue introduced earlier in our model can be locally generated by mechanosensitive self-organization and does not require the existence of pre-patterned polarity cues. To generate a large-scale pattern from locally generated anisotropic cues, they need to be aligned in neighboring regions. This local alignment is captured in *Equation 8* by the orientation coupling term with strength *D*, which is similar to alignment terms found in anisotropic physical systems, such as liquid crystals (*Gennes and Prost, 1993*; *Jülicher et al., 2018*; *Marchetti et al., 2013*).

To discuss cell morphology profiles in the wing disc, we consider a simplified tissue model with radial symmetry, where the rate of radial cell rearrangement $R_{rr}$ is given (as estimated in Appendix 1 part 2) and the cell shape pattern and tissue stress pattern are calculated. Using a fit of cell elongation to the experimental data, we find a set of parameter values that accounts for the observed cell elongation patterns in the wing disc (*Table 1*, Appendix 1, *Figure 6C*). From this cell elongation pattern also follows the cell area pattern (as described above, *Figure 3*).

We conclude that our mechanosensitive model can account for the radial pattern of cell morphology in the wing disc. Due to the relatively large number of parameters used to fit a single experimental curve, there are large uncertainties when estimating parameter values. Nonetheless, these uncertainties do not affect the qualitative prediction that a reduction in mechanosensitivity $\mu$ would lead to less polarization and thereby reduced cell elongation (see Appendix 1 part 4). We next test this prediction of our model experimentally.

## Suppression of mechanosensitivity weakens the gradients in cell elongation and cell size

In order to test our prediction that the reduction of mechanosensitivity will reduce the magnitude of cell elongation, we used RNAi to reduce the levels of Myosin VI (MyoVI), a molecular motor

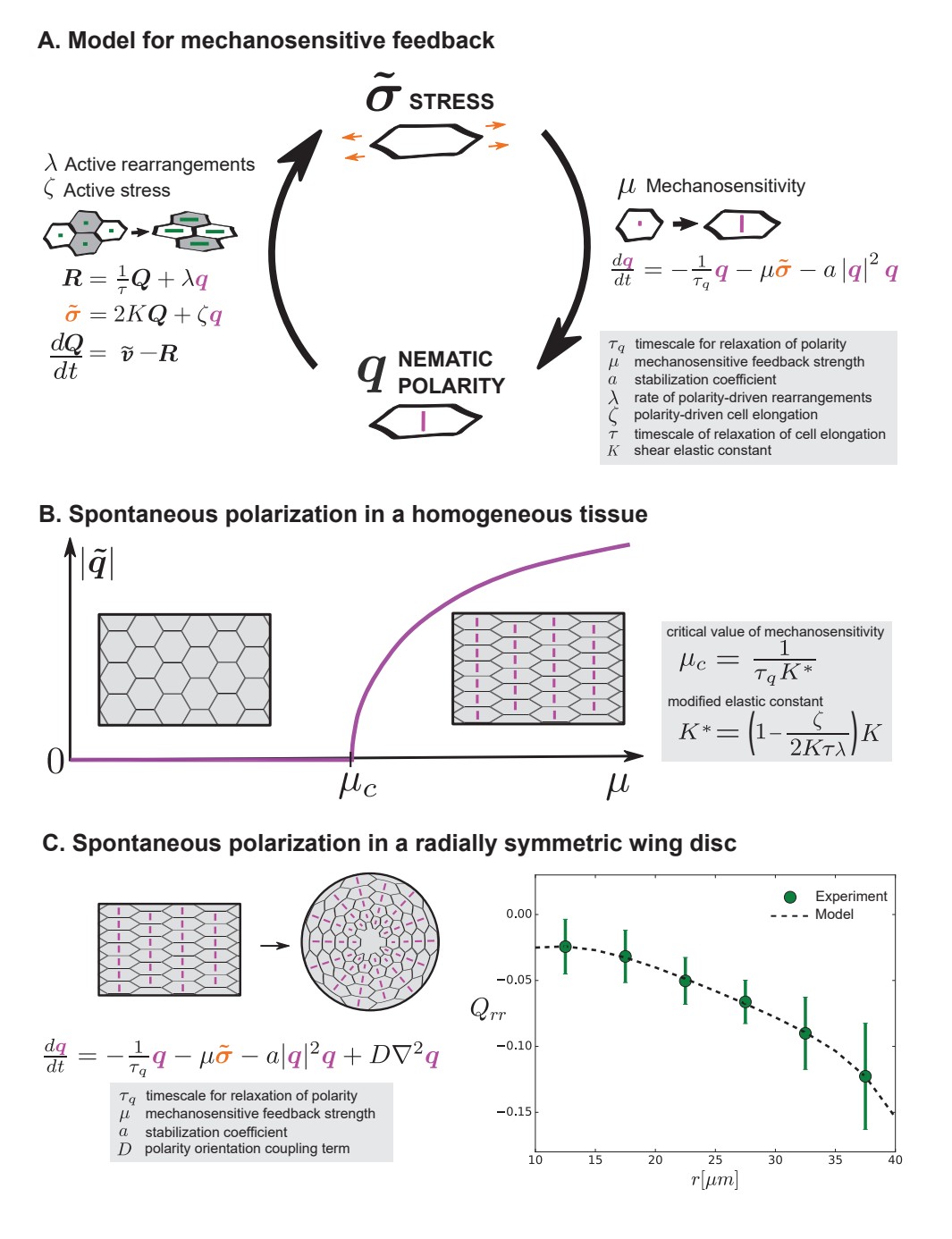

**Figure 6.** Mechanosensitive feedback generates self-organized patterns of cell morphology. (**A**) We introduce mechanosensitive feedback into our model with a dynamic equation relating the orientational cue $q$ to tissue stress. With a change in $q$, there can be a change in polarity-driven cell rearrangements and stress. (**B**) Above a critical value of $\mu$, the isotropic state is unstable and the system spontaneously polarizes. (**C**) Application of the model to the radially symmetric wing disc. Generation of large-scale tissue polarity requires the orientation coupling term $D$. We find a set of parameters that can account for the experimental data on cell elongation in the wing disc (see Appendix 1 part 4 and **Table 1**).

**Table 1.** Parameter values obtained by fitting the tangential cell elongation profile shown in *Figure 6C* to the self-organized model.

Values in the last two rows are boundary conditions that are required to calculate the tangential cell elongation profile. The reported parameter uncertainty intervals were obtained by fitting 101 uniformly sampled tangential cell elongation profiles from the range defined by the standard deviation in the experimental tangential cell elongation profile (error bars in *Figure 6C*). The interval limits are the 10$^{th}$ and 90$^{th}$ percentiles of the obtained values.

| Fit parameter | Parameter value | Uncertainty interval |
|---|---|---|
| $q_0^2$ | 0.01 | $[-0.06, 0.11]$ |
| $\lambda \tau$ | 1.0 | $[0.2, 1.3]$ |
| $\frac{D}{\alpha}$ | $0.5 \mu m^2$ | $[0.2, \; 27] \; \mu m^2$ |
| $\frac{1}{D}\left(\frac{1}{\tau\lambda} + 2K\mu\tau\right)R_{rr}$ | $9 \cdot 10^{-4} \mu m^{-2}$ | $[0.0, 3.2] \cdot 10^{-3} \mu m^{-2}$ |
| $q_{rr}(r_{in})$ | 0.024 | $[0.014, 0.105]$ |
| $\partial_r q_{rr}(r_{in})$ | $-1.2 \cdot 10^{-3} \mu m^{-1}$ | $[-9.3, 0.0] \cdot 10^{-3} \mu m^{-1}$ |

implicated in mechanosignaling. MyoVI, encoded by jaguar (*jar*) in *Drosophila*, is an upstream component of a Rho-dependent signaling pathway that reorganizes the actin-myosin cytoskeleton in response to mechanical stress (*Acharya et al., 2018*). Experiments in wing discs also indicate that mechanosensation involves Rho polarization and signaling (*Duda et al., 2019*). We performed RNAi targeting MyoVI in the wing pouch using *nub-Gal4* and evaluated cell morphology at the end of larval development ($\sim 120hr$ AEL). We observe a clear reduction in the magnitude of tangential cell elongation as compared to wild type at this stage (*Figure 7A–D*, *Figure 7—figure supplement 1*). In addition, our model predicts that such reduction of cell elongation would result in an increase of cell area in the wing (*Equations 1-6*). The observed pattern of increased cell area upon reducing MyoVI levels with RNAi is consistent with this prediction (*Figure 7E–F*). Therefore, the qualitative predictions of our model upon reducing the mechanosensitive feedback strength $\mu$ are confirmed by the experimental downregulation of the mechanosensitive motor MyoVI.

## Discussion

Here, we have shown that patterns of cell shape and stress in the mid-third instar *Drosophila* wing disc do not rely on PCP pathways or differential growth. Instead, radially-oriented T1 transitions and tangential cell elongation emerge via mechanosensitive feedback in a self-organized process. We have presented a continuum model of tissue dynamics for this self-organization based on a mechanosensitive nematic cell polarity that accounts for the observed patterns of cell area, T1 transitions, and cell shape. Our work highlights a mechanism for the self-organized emergence of cellular patterns in morphogenesis, expanding our understanding of pattern formation emerging from mechanical feedbacks in active systems (*Bois et al., 2011*; *Howard et al., 2011*; *Recho et al., 2019*).

### A pattern of T1 transitions is critical for cell morphology patterning in the *Drosophila* wing

Our work shows that the spatial pattern of T1 transitions is an integral part of the emergence of tissue organization during wing development. In contrast to situations such as germband extension, where T1 transitions exhibit clearly discernible patterns, the patterns of T1 transitions in the wing disc have been elusive. Many T1 transitions occur in the tissue in seemingly random orientations. However, on average, they exhibit a spatial pattern. We revealed these patterns by quantifying the nematics of T1 transitions and cell shape changes using the previously-described triangle method (*Merkel et al., 2017*) and then quantified them with radial averaging (*Figure 2A–B*). In this way, we revealed that a radial pattern of T1 transitions is linked to a tangential pattern of cell elongation.

Given this radial pattern of T1 cell rearrangements, the observed cell morphology pattern follows from a continuum tissue model based on a radially oriented nematic cell polarity field (*Figure 3*). The polarity-oriented radial T1s create a cell shape pattern with corresponding patterns of tissue

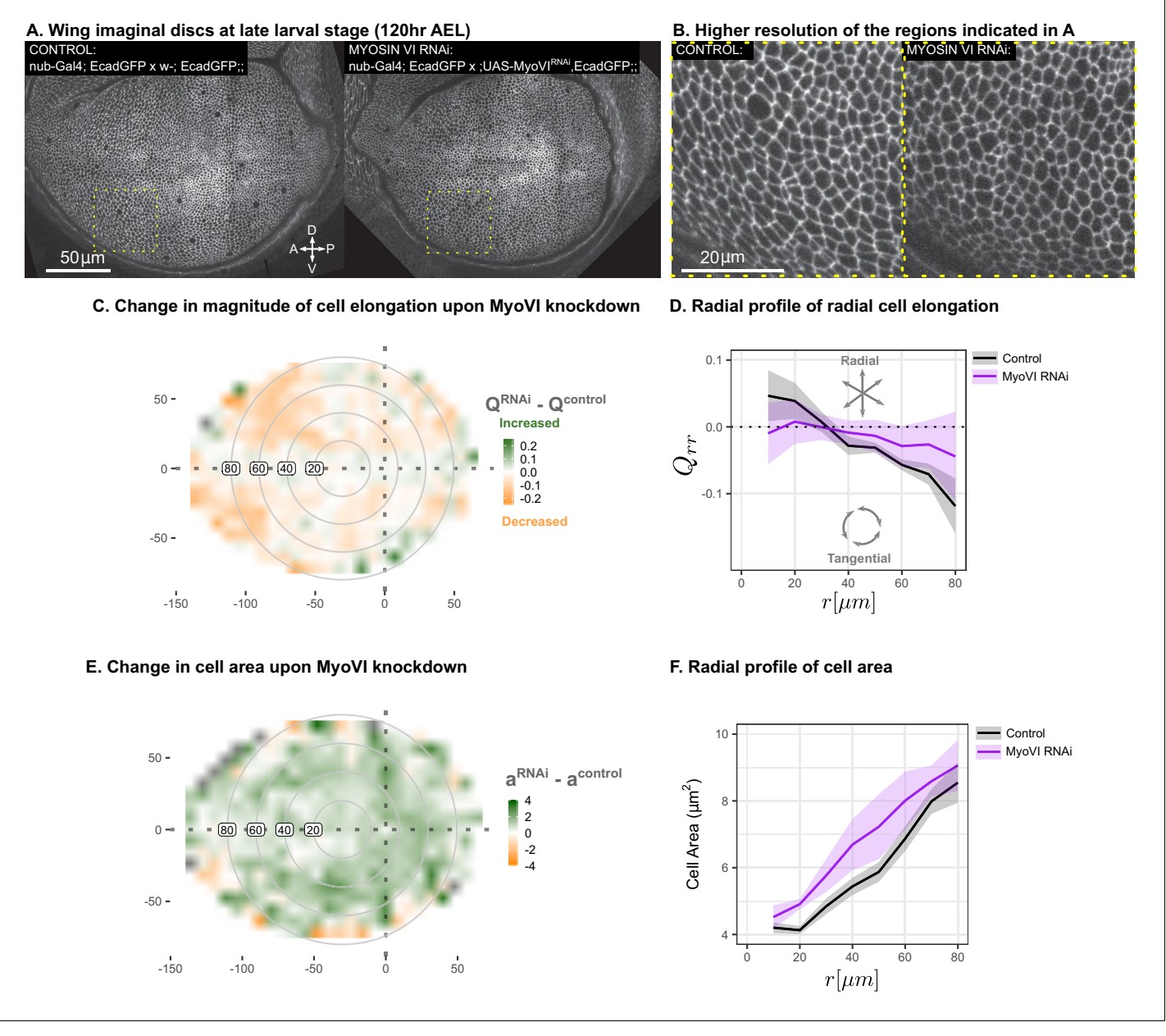

**Figure 7.** Suppression of mechanosensitivity weakens the gradients in cell elongation and cell size. MyoVI levels were reduced with RNAi in the pouch (*nub-Gal4 > UAS-MyoVI*$^{RNAi}$), and cell elongation and cell area were assessed at the end of larval development (119*hr* AEL). Data are included in *Figure 7—source data 1* and *2*. Shown in (**A**) are representative images, with apical cell boundaries marked by Ecadherin-GFP. Yellow box indicates the inset that is presented in higher resolution in (**B**). The spatial pattern of radial cell elongation is presented in *Figure 7—figure supplement 1*. The difference in magnitude of cell elongation (**C**) or cell area (**E**) between *MyoVI*$^{RNAi}$ and the corresponding control is presented. Axis labels indicate the distance to the AP boundary (X) or DV boundary (Y) in *μm*. Gray circles indicate radial bins, with numbers corresponding to distance (in *μm*) from the center. (**D,F**) The radial profile in the radial component of cell elongation $Q_{rr}$ (**D**) or cell area (**F**) was quantified for each genotype. Plots in (**C-F**) represent averages of $N = 6 - 7$ wing discs per genotype.

The online version of this article includes the following source data and figure supplement(s) for figure 7:

**Source data 1.** Triangle elongation for MyoVI RNAi.

**Source data 2.** Cell area for MyoVI RNAi.

**Figure supplement 1.** Spatial pattern of cell elongation upon MyoVI knockdown.

stress and tissue area pressure. The 2D area pressure is higher in the center and is lower toward the periphery. Note that this pressure profile does not rely on differential proliferation, as was previously proposed (*Mao et al., 2013*) but instead relies on a radial pattern of T1 transitions. We test our model using a novel circular laser ablation method. This method allows us to determine specific combinations of tissue parameters. In particular, we estimate the ratio of elastic constants $2K/\overline{K} = 3.4 \pm 0.4$ and $K^*/K = 0.05 \pm 0.02$, as well as the cell shape relaxation timescale $\tau = 2 \pm 2hr$.

This analysis raised the question of which nematic cell polarity cues guide the cell rearrangement and cell elongation patterns. PCP pathways are required for the proper orientation of T1 transitions in other contexts (*Bosveld et al., 2012*). However, we found that neither of the two known PCP pathways in the wing are required for the observed tangential cell elongation patterns (*Figure 5*). We instead show that an orientation cue can arise through self-organization via mechanosensitivity and identify MyoVI as a key molecular player.

## Mechanosensitive feedback can create self-organized patterns of cell morphology

Cell polarity cues can emerge via mechanosensitive feedback by transforming mechanical cues into chemical anisotropies. Nematic cell polarity can then orient active stresses and thereby amplify the mechanical stimulus (*Figure 6*). We introduce this mechanosensitive feedback in our continuum theory, which quantitatively describes the emergence of patterns of cell shape and cell rearrangements. The strength of this mechanosensitive feedback is described by a parameter $\mu$. If $\mu$ exceeds a critical value $\mu_c$, an orientation cue and elongated cell shapes spontaneously emerge (*Figure 6B*). This model can account for the observed patterns of cell area and cell elongation in the wing disc and predicts that the reduction of mechanosensitivity will result in reduced cell elongation. To test this prediction, we perturbed a RhoA-dependent mechanotransduction pathway by lowering levels of an upstream component, MyoVI, using RNAi (*Figure 7*). We find a clear phenotype of reduced cell elongation and increased cell areas in the center region, as predicted by our model. In the spatially resolved model, which includes the term coupling polarity orientation of neighboring cells (*Figure 6C*), a polarity pattern can be induced even below $\mu_c$ by imposing polarity at the tissue boundaries. Thus, the residual pattern of cell elongation that we observe after removal of MyoVI in the wing pouch could be due to polarity existing outside of this perturbed region. In addition, the residual pattern could also indicate an incomplete knockdown from RNAi or the presence of other mechanosensitive elements in the tissue. Nevertheless, given the clear phenotype that is fully consistent with our model, we propose that the mechanosensitive feedback mechanism is a significant determinant of the cell shape patterns in the wing pouch.

Our data, together with the fact that MyoVI is involved in Rho-dependent activation of actin-myosin cytoskeleton (*Acharya et al., 2018*), suggest that MyoVI is a molecular component of the mechanosensitive feedback we describe in our self-organized model. However, the molecular nature of the cue that defines the nematic cell polarity is unknown. This cue may organize the structure or dynamics of the actin-myosin cytoskeleton or the actin-myosin system itself could define nematic cell polarity. Indeed, it has been shown that Myosin II (MyoII) localizes to long cell boundaries in the wing (*Legoff et al., 2013*), corresponding to a nematic polarity aligned with the nematic cell polarity $q$. Also, wing disc stretching experiments have shown that MyoII can polarize in response to exogenous stress in a Rho-dependent manner (*Duda et al., 2019*). Furthermore, it has been suggested that MyoII polarity arises as a consequence of cell stretching and functions as a negative mechanical feedback (*Legoff et al., 2013*), consistent with the role of $q$ in our model. Precisely how the actin-myosin cytoskeleton is affected by MyoVI in this system and how these cytoskeletal elements together guide cell rearrangements in response to anisotropic tissue stresses and cell shape changes remain open questions for future research.

Lastly, our laser ablation analysis shows that the region around the DV boundary has a different ratio of elastic constants than the rest of the tissue, which could affect the self-organized pattern formation we describe. Therefore, it will be interesting to study how Wingless/Notch signaling, which defines the DV boundary, may influence the mechanical properties that lead to mechanosensitive self-organization of polarity and morphology. In addition, we observe a richer pattern emerging very late in development (see *Figure 7—figure supplement 1*, *Figure 5*), including a region anterior to the AP boundary that is radially elongated. Future research will expand upon the model presented here to explore the dynamics of these patterns.

In summary, we used the *Drosophila* wing disc to identify a mechanism by which tissue morphology can arise from the self-organization of a mechanical feedback coupling cell polarity to active cell rearrangements. This mechanism is general and could be employed in other tissues and organisms to generate patterns of cell shape and cell area. Thus, we hope our work inspires new avenues of research that integrate theory and experiment to understand biological self-organization.

# Materials and methods

## Key resources table

| Reagent type (species) or resource | Designation | Source or reference | Identifiers | Additional information |
|---|---|---|---|---|
| Antibody | Anti-*Drosophila* SRF (Mouse monoclonal) | Active Motif | Cat#:39093; RRID:AB_2793614 | lot# 03504001; IF(1:100) |
| Antibody | Anti-GFP (Rabbit polyclonal) | Invitrogen | Cat#:A11122; RRID:AB_221569 | IF(1:1000) |
| Antibody | Anti-*Drosophila* Patched; Apa 1, conc; (Mouse monoclonal) | Developmental Studies Hybridoma Bank | Cat#:Apa1; RRID:AB_528441 | IF(1:100) |
| Antibody | Anti *Drosophila* Wingless; 4D4, conc; (Mouse monoclonal) | Developmental Studies Hybridoma Bank | Cat#:4d4; RRID:AB_528512 | IF(1:100) |
| Antibody | Anti-Dachsous, 2828; (Mouse monoclonal) | *Merkel et al., 2014* | | IF(1:50) |
| Chemical compound, drug | Grace's insect medium | Sigma Aldrich | Cat#:G9771−10 × 1L | |
| Chemical compound, drug | Fetal Bovine Serum | Thermo-Fisher | Cat#:10270098 | |
| Chemical compound, drug | 20-Hydroxyecdysone | Sigma Aldrich | H5142 | |
| Chemical compound, drug | Penicillin-Streptomycin | Sigma Aldrich | P4333 | |
| Genetic reagent (*D. melanogaster*) | Ecadherin-GFP; Ecad-GFP | *Huang et al., 2009* | | yw-; EcadGFP;; |
| Genetic reagent (*D. melanogaster*) | nub-Gal4 | Bloomington *Drosophila* Stock Center | Cat#:86108; RRID:BDSC_86108 | Combined with EcadGFP in this work |
| Genetic reagent (*D. melanogaster*) | fat+dachs$^{RNAi}$ | Vienna *Drosophila* Resource Center | Cat#:9396; Cat#:12555 | Combined with Dcr2 on 1st chromosome and EcadGFP on 2nd in this work |
| Genetic reagent (*D. melanogaster*) | pk30 | Bloomington *Drosophila* Stock Center | Cat#44229; RRID:BDSC_44229 | Combined with EcadGFP in this work |
| Genetic reagent (*D. melanogaster*) | MyoVI$^{RNAi}$ | Vienna *Drosophila* Resource Center | Cat#37534 | Combined with EcadGFP in this work |
| Software, algorithm | TissueMiner | *Etournay et al., 2016* | | |

### Lead contact and materials availability

This study did not generate any new unique reagents. All requests for further information and reagents may be directed to the lead author, natalie_anne.dye@tu-dresden.de.

### Experimental model and subject details

All experiments were performed with *Drosophila melanogaster*, using lines that are publicly available and previously published. Our *Drosophila* lines were fed with a standard media containing cornmeal, molasses agar and yeast extract and grown under a $12hr$ light/dark cycle. All experiments were performed at $25°C$. Both males and females were analyzed, and the sex of the animals was not recorded, as we have no reason to believe there is any sexual dimorphism in the studied phenomenon. To synchronize development, we collected eggs deposited within a defined time window on apple juice agar plates. To do so, we transferred the flies from standard food vials to cages covered by apple juice agar plates containing a dollop of yeast paste for food. After at least $2hr$, the plates were replaced, and the timing of collection started. Eggs laid within a $\sim 2hr$ time window were

collected by cutting out a piece of the agar and transferring it to a standard food vial. We limited the number of eggs per vial to <15 to avoid crowding. The middle of the time window for egg collection was considered to be $0hr$ AEL. Experiments from timelapse imaging and laser ablation (**Figures 1**, **2,** and **4**) were captured after explanting at $96hr$ AEL, whereas those involving RNAi (**Figures 5** and **7**) were explanted at $110 - 120hr$ AEL to allow the maximal amount of time for the RNAi phenotype to emerge. The specific genotypes used for each experiment are indicated in the Key Resources Table and in the figures.

## Method details

### Timelapse image acquisition and processing

#### Sample preparation

Wing explants were grown *ex vivo* as described previously (**Dye et al., 2017**). Briefly, wing discs were dissected from larvae in growth media (Grace's cell culture media + 5% fetal bovine serum $+20nM$ 20-hydroxyecdysone + Penicillin-Streptomycin) at room temperature. Then, they were transferred to a Mattek #1 glass bottom petri-dish to the center of a hole cut in a double-sided tape spacer (Tesa 5338) and covered with a porous filter. The dish was then filled with fresh growth media.

#### Acquisition

Data from movies 1 to 3 were used previously (**Dye et al., 2017**). Movie 4 was acquired after publication of the first manuscript, in the same exact way as movies 1 to 3. Briefly, Ecadherin-GFP-expressing wing discs were imaged in growth media using a Zeiss spinning-disc microscope to acquire $0.5\mu m$ spaced Z-stacks at 5 min intervals with a Zeiss C-Apochromat 63X/1.2NA water immersion objective and $2 \times 2$ tiling (10% overlap). This microscope consisted of an AxioObserver inverted stand, motorized xyz stage, stage-top incubator with temperature control set to $25°C$, a Yokogawa CSU-X1 scanhead, a Zeiss AxioCam MRm Monochrome CCD camera (set with $2 \times 2$ binning), and 488 laser illumination. We circulated growth media during imaging using a PHD Ultra pump (Harvard Apparatus) at a rate of $0.03ml/min$. Time $0hr$ of the movie was considered to be the start of imaging acquisition, typically $45 - 60min$ from the start of dissection (time required for sample and microscope preparation).

Movie 5 was acquired on a newer microscope with a larger field of view that eliminated the need to tile across the wing pouch. This microscope has an Andor IX 83 inverted stand, motorized xyz stage with a Prior ProScan III NanoScanZ z-focus device, a Yokogawa CSU-W1 with Borealis upgrade, and a Pecon cage incubator for temperature control at $25°C$. We used an Olympus 60x/1.3NA UPlanSApo Silicone-immersion objective with 488 laser illumination and an Andor iXon Ultra 888 Monochrome EMCCD camera. We acquired $0.5\mu m$ spaced Z-stacks of a single tile at $5min$ intervals, as for the other movies. Movie 5 was also acquired without the constant flow of new media.

For all movies, care was taken to limit light exposure, using laser power values of $<0.08mW$ and exposure times less than $350ms$ per image.

#### Processing

Raw Z-stacks were denoised using a frequency bandpass filter and background subtraction tools available in FIJI (**Schindelin et al., 2012**). Then, we used a custom algorithm as described previously (**Dye et al., 2017**) to make 2D projections of the apical surface, marked by Ecadherin-GFP. This algorithm also outputs a height-map image, in which the value for each pixel corresponds to the level in the Z-stack of the identified apical surface. For those movies that were tiled, we used the Grid/Collection Stitching FIJI plugin (**Preibisch et al., 2009**) to stitch the 2D projections, and then used that calculated transformation to stitch the height map images, so that we could correct the cell area and elongation values for local curvature (see below). We focus in this work exclusively on the disc proper layer, which is the proliferating layer of the disc that goes on to produce the adult wing. We did not consider cell shape patterns in the peripodial layer, the non-proliferating layer of the wing disc that is largely destroyed at the onset of pupariation.

## Single timepoint image acquisition and processing for analysis of RNAi/mutant phenotypes

### Sample preparation

Samples for single timepoint imaging were acquired exactly as described for live imaging: dissected in growth media and imaged in Mattek dishes under a porous filter.

### Acquisition

All imaging of single timepoint data was performed with the same microscope described above for Movie 5. We did not acquire timelapse data for these genetic perturbations, and thus we chose to image the entire disc using $2 \times 2$ tiling, $0.5 \mu m$ spaced Z-stacks. Approximately $6 - 10$ discs were imaged per dish, within approximately $30 - 60 min$ .

### Processing

Tiled images were stitched together as Z-stacks; then we obtained the apical surface projection and its corresponding height map as described above and in *Dye et al., 2017*.

## Circular laser ablation

### Acquisition

Wing discs from mid-third-instar larvae ($96 hr$ AEL) were dissected and mounted exactly as was done for the live imaging timelapses. Due to constraints on the speed of ablation, we only cut regions of the wing disc pouch that were completely flat, so that we could cut in a single plane at the apical surface (rather than having to cut in each plane of a Z stack). Where this flat region lies depends on how the disc happens to fall on the coverslip during mounting. Because the anterior compartment is larger and higher, most of our ablations are in this compartment. To access other regions, we also mounted some wing discs on an agarose shelf: stripes of 1% agarose (in water) were dried onto the surface of the coverslip and wing discs were arranged their anterior half propped up on the agarose shelf prior to adding the porous filter cover.

Ablations were performed using ultraviolet laser microdissection as described in *Grill et al., 2001* using a Zeiss 63X water objective. First, we took a full Z-stack of the sample prior to the cut. Then, we selected a $7 \mu m$ radius circle that would ablate in the flattest region of the tissue. No imaging is possible during ablation, but we acquired a $2 min$ timelapse immediately after the cut in a single Z-plane. This timelapse data was not used except to estimate whether or not the sample was fully cut. After, once the sample has finished expanding but not started to heal ($\sim 2 min$), we took another full Z-stack to image the endpoint. We excluded a small number of data points if any of the following were true: (1) the inner piece remaining after the ablation was no longer visible (sometimes it floats or is destroyed); (2) the cut appeared to expand highly asymmetrically (rare); (3) the wing discs were clearly too young to be considered $96 hr$ AEL (poor staging).

### Immunofluorescence after ablation

Due to a limited field of view on the microscope used for ablation, we performed immunofluorescence after the ablation in order to better estimate the position of the cut in the wing pouch. After all the discs in the dish were ablated (<10 discs/dish), the entire dish was fixed through the filter by adding 4% PFA and incubating $20 min$ at room temperature. After, the dish was rinsed and kept in PBST (PBS + 0.5% Triton X-100) until all discs from that image acquisition day were completed ($2 - 4 hr$). All samples were then blocked using 1% BSA in PBST + $250 mM$ NaCl for $45 min - 1 hr$, and then incubated in primary antibody overnight (diluted in BBX: 1% BSA in PBST). Initially, we labeled samples with SRF (Active Motif, 1:100 dilution), but later we switched to Patched and Wingless (DSHB, 1:100) to identify the compartment boundaries. Primary antibody toward GFP (recognizing the Ecadherin-GFP) was also included at 1:1000. After overnight incubation at $4°C$, we washed with BBX, followed by BBX+ 4% normal goat serum (NGS), for at least $1 hr$. Secondary antibodies were added for $2 hr$ at RT in BBX+NGS. Finally, samples were washed $4 x 10 min$ in PBST and imaged in this media. Imaging was performed on one of the two spinning disc microscopes described above for live imaging. We matched the stained samples with the ablation images by (1) keeping track of the position of each disc on the dish and where it was ablated during acquisition and (2) morphology of the disc before/after ablation. All antibody information are listed in the Key Resources Table.

## Validation of Fat/D PCP RNAi

### Sample preparation

To immunostain larval wing discs, late third instar larvae were partially dissected in $1\times$ PBS, inverting the heads to expose the wing discs, and then fixed in 4% PFA for $20min$. Samples were then washed twice for $15min$ each with PBX2 ($1\times$ PBS + 0.05% Triton X-100), and blocked in BBX250 (PBX2 + $1mg/ml$ BSA + $5mM$ NaCl) for $45min$. Incubation in primary antibody diluted 1:50 in BBX (PBX2 + $1mg/ml$ BSA) was performed overnight at $4°C$. We used a mouse monoclonal antibody against Dachsous (**Merkel et al., 2014**). After, the sample was washed twice with BBX, $20min$ for each wash, followed by one $45min$ wash with BBX + $4°C$ normal goat serum (NGS). Incubation with secondary antibody was performed either for $2-3hr$ at room temperature or overnight at $4°C$. We used a goat anti-mouse Alexa-Fluor 647 as the secondary antibody (ThermoFisher,Cat No. A28181) at 1:500 dilution in (BBX+4% NGS). The secondary antibody was removed by washing twice with PBX2, followed by washing twice with $1\times$ PBS. The tissues were stored in PBS and dissected from the body wall just before mounting. Wing discs were mounted on a slide within a thin channel created by two strips of double-sided tape (Tesa 5338). Wings were transferred to the middle of the channel with a p200 pipette, the excess PBS was removed, and the tissues were arranged with apical sides up. Next, the sample was covered with a $22\times22mm$ #1 coverslip, adhering the coverslip to the double-sided tape. Vectashield mounting medium (Vector laboratories) was added to one side of the coverslip and allowed to seep in by capillary action. Excess Vectashield was removed, and the sides of the coverslip were sealed using transparent nail polish. The slides were then stored at $4°C$ until imaging.

To image the adult wing morphology, male flies of the desired genotype were collected and stored in isopropanol. Wings were dissected and collected in isopropanol. To mount, wings were transferred to a glass slide using a p200 pipet. In a centrifuge tube, $500\mu l$ of isopropanol was mixed with $500\mu l$ of Euperal, and about $50\mu l$ of this mix was added to the slide containing the wings. The wings were then aligned on the slide, adding more isopropanol/Euperal mix when necessary to avoid drying. Once the wings were all aligned, more isopropanol-Euperal mix was added and allowed to partially dry. Once the mix was almost dry, a clean $22\times22$ coverslip containing about $20-50\mu l$ of Euperal was inverted on top of the wings. The slides was allowed to cure for $24hr$ before imaging.

### Acquisition

Immunostained samples were imaged on an Olympus IX81 microscope equipped with a spinning disk module (Yokogawa) and back illuminated EMCCD (Andor Technology, iXon Ultra 888) with an exposure time of $100ms$ and EM gain of 250. A confocal z-stack of immunostained samples was acquired using a $60\times$ silicon immersion objective with $0.47\mu m$ z-spacing.

Adult wings were imaged on an inverted wide field microscope (Zeiss) using a $5\times$ objective. The images were analyzed using a custom code written in MATLAB (Mathworks, USA) to measure wing area.

## Quantification and statistical analysis

### Cell segmentation, tracking, and alignment

#### Segmentation and tracking

Using the 2D projections, we performed cell segmentation and tracking using the FIJI plugin, TissueAnalyzer (**Aigouy et al., 2016**). We manually corrected errors in the automated segmentation and tracking as much as possible and then generated a relational database using TissueMiner (**Etournay et al., 2016**). Images were rotated to a common orientation (Anterior left, dorsal up). We then manually identified three regions of interest at the last point in the timelapse, using the FIJI macros included with TissueMiner: the 'blade' was defined roughly as an elliptical region surrounded by the most distal folds; and the Anterior-Posterior and Dorsal-Ventral boundaries were estimated using the brightness of Ecadherin-GFP and apical cell size (**Jaiswal et al., 2006**).

For the timelapse data, we used only the cells within these manually defined regions that were trackable during the entire course of the movies. An example of this region is presented for Movie 1 in **Figure 1A**. Furthermore, we also excluded from all of our analysis the first $2hr$ of imaging, the so-called adaption phase, where cells uniformly shrink in response to culture (**Dye et al., 2017**).

For the RNAi data (**Figures 5** and **7**), we did not acquire timelapses, rather a single timepoint at late stages of development. We nevertheless chose to analyze these data using the same Tissue-Miner workflow for simplification. Because TissueMiner was developed for timelapse data, however, it requires at least two timepoints. Thus, we duplicated the data and labeled the two images as if they were timepoints 0 and 1 of a timelapse. We then only analyzed timepoint 0. The regions of interest in these data, therefore, are manually defined and not simply the region that is trackable (since it is static data).

## Alignment on compartment boundaries

To average across all movies of timelapse data, or all discs of the same genotype of the static RNAi data, we generated a disc-coordinate system by normalizing the position of each cell to the AP and DV boundaries for that disc. To do so, we averaged the absolute xy positions of all the cells in the AP or DV boundaries over all time after the adaption phase. We then calculated the distance of each cell to the new X axis (average position of the AP boundary) and Y axis (average position of the DV boundary).

## Analysis of cell size and shape

### Definition of the cell elongation tensor

The cell elongation tensor $Q$ is a traceless symmetric tensor that quantifies the anisotropy of cell shapes in a region of the tissue. We define cell elongation using a triangulation of the tissue obtained by connecting centroids of connected cells (**Etournay et al., 2015**; **Merkel et al., 2017**). The state of each triangle is described by a tensor $s$, defined by a mapping an equilateral reference triangle to the triangles in the tissue. The state tensor contains information about the triangle elongation tensor $\tilde{q}^t$, orientation angle $\theta$, and area $a$:

$$s = \left(\frac{a^t}{a_0^t}\right)^{1/2} \exp(\tilde{q}^t) R(\theta).$$

Here, $a_0^t$ is the area of the reference triangle, and $R$ is a 2-dimensional rotation matrix. Cell elongation in the tissue region is defined as an area weighted average of triangle elongation. For details about the method see **Merkel et al., 2017**.

### Adjustment of cell area and elongation to account for tissue curvature

The wing disc pouch has a slightly domed shape (**Figure 1—figure supplement 1A**). After projection onto a 2D surface, the cell shapes and areas will be distorted. To ensure that the radial profiles we measure in a 2D projection (**Figure 1B,C,E and F** and **Figure 3F,G**) are not a result of tissue curvature, we account for the distortion caused by projection. We use the height maps generated by our projection algorithm, which identifies the apical surface of the pouch within the 3D Z-stacks. We smooth this height map with a gaussian filter of width $\sigma = 2~\mu m$ to find the height field $h(x, y)$, and then we calculate the height gradient field. We then smooth the result again with the same gaussian filter to find $\vec{\nabla} h = (\partial_x h, \partial_y h)$. The deprojected cell or triangle area $a_0$ is obtained from the measured area $a$ as $a_0 = a\sqrt{1 + |\vec{\nabla} h|}$, where $\vec{\nabla} h$ is evaluated at the cell center. To find the deprojected cell elongation of a triangle we evaluate $\vec{\nabla} h$ at the triangle center. Then, we find the angle of steepest ascent $\alpha = \arctan(\partial_y h / \partial_x h)$ in the projected plane and define the tilt transformation:

$$N = \begin{pmatrix} \frac{\sqrt{1+|\vec{\nabla} h|}+1}{2} + \frac{\sqrt{1+|\vec{\nabla} h|}-1}{2}\cos(2\alpha) & \frac{\sqrt{1+|\vec{\nabla} h|}-1}{2}\sin(2\alpha) \\ \frac{\sqrt{1+|\vec{\nabla} h|}-1}{2}\sin(2\alpha) & \frac{\sqrt{1+|\vec{\nabla} h|}+1}{2} - \frac{\sqrt{1+|\vec{\nabla} h|}-1}{2}\cos(2\alpha) \end{pmatrix}$$

We apply this transformation to the triangle shape tensor $s$, as defined in **Merkel et al., 2017**, to determine the deprojected triangle shape tensor $s_0$:

$$s_0 = Ns$$

The cell elongation tensor $\mathbf{Q}$ is obtained from the triangle shape tensor $\mathbf{s}_0$ as the corresponding area weighted average using the deprojected cell area, as described in *Merkel et al., 2017*.

## Spatial maps of cell size and shape

To generate the color-coded smoothed plots of area and elongation (*Figure 1B–C*), we divided the aligned wings into a grid with boxsize = 10 pixels ($\sim 2\mu m$). For each position, we averaged cell area or performed an area-weighted averaged of triangle elongation in a neighborhood box = 20 pixels ($\sim 4\mu m$). To create the nematic elongation pattern, we similarly averaged elongation of triangles whose centers lie within each grid box, with grid box size = 30 pixels ($\sim 6\mu m$). Tables containing the corrected cell area and triangle elongation values for each movie at each timepoint have be uploaded to Dryad: doi:10.5061/dryad.jsxksn06b.

## Calculation of radial elongation center point

We define the center of the wing pouch to be on the DV boundary. To determine the location of the center along the DV boundary $x_c$ we divide the pouch into four regions defined by the DV boundary and a line perpendicular to it located at some position $x$, as shown in *Figure 1—figure supplement 1E*. Then, we define a function:

$$F(x) = A_I \langle Q_{xy} \rangle_a^I - A_{II} \langle Q_{xy} \rangle_a^{II} + A_{III} \langle Q_{xy} \rangle_a^{III} - A_{IV} \langle Q_{xy} \rangle_a^{IV}$$

for all values of $x$ along the DV boundary. Here, $A^{I-IV}$ are the areas of the four regions and $\langle Q_{xy} \rangle_a^{I-IV}$ are the area weighted averages of the cell elongation component $Q_{xy}$ calculated in the four regions. Finally, $x_c$ is the value that minimizes $F(x)$ (*Figure 1—figure supplement 1F*). We find that the center point lies just anterior to the intersection with the anterior-posterior (AP) boundary (*Figure 1*, *Figure 1—figure supplement 1*, consistent with *Legoff et al., 2013*).

In time-lapse experiments, cell centers were determined using the last 20 timepoints. In single timepoint image experiments (*Figures 5* and *7*), all images of a single genotype were used together after alignment on the DV and AP boundaries.

## Exclusion of the band of cells near the DV boundary

In contrast to the other regions (blade, AP and DV boundaries), the definition of the band around the DV boundary (*Figure 1—figure supplement 2*) and the region of the blade that excludes this region (*Figure 1D*) were not defined manually using the FIJI scripts of TissueMiner. Rather, we defined them after the TissueMiner databases were generated, using the position of cells relative to the DV boundary in the last frame. Using the plots of the radial elongation on each disc, we estimated the width of the stripe region and then filtered for cells included in and excluded from this region. Once these cells in the last frame were identified, we used the UserRoiTracking.R of TissueMiner to backtrack these two regions, producing a list of all cells belonging to this lineage that are traceable forward and backward in the movie.

For the static RNAi data, we used the average width of the band around the DV boundary from timelapse data as an estimate and excluded this region from the static images to analyze the radial gradients in cell area and elongation (*Figures 5* and *7*).

## Plotting area and elongation versus distance to the center of elongation

For the timelapse data, we defined the radial elongation center (described above) for each movie and then calculated the distance away from this center for each cell. After excluding the band of cells around the DV boundary, we binned cells by radius by rounding to the nearest $10\mu m$. We also binned the movie across five equal time windows ($\sim 2hr$), excluding the adaption phase. Average cell area and area-weighted average of cell elongation were calculated for each of these bins in each time group for each movie. Including data from all five movies, we report the global average and its standard deviation (*Figure 1E–F*). For *Figure 3*, we averaged over the last $\sim 5hr$. For the band of cells around the DV boundary (*Figure 1—figure supplement 2*), we binned cells in $x$, rather than in radius, and report the gradient along the $x$ axis, defined as the DV boundary.

The static RNAi data were analyzed similarly. We report the global average of all discs in the genotype and the standard deviation (*Figures 5D,I* and *7D*). The number of discs analyzed in each genotype is listed in the figure legend.

## Regional analysis of isotropic tissue deformation

### Spatial maps of cellular contributions to isotropic tissue deformation

To analyze the pattern of isotropic deformation, we locally averaged cell behavior by dividing the tissue into a grid centered upon the crossing of the AP and DV boundaries. First, we determined the average position of cells in the AP and DV boundaries to generate a common frame of reference across all movies. Second, we divided the tissue visible in the last frame into a grid centered on these compartment boundary positions, with grid size = $8 \mu m$. Grid boxes that were incompletely filled (less than 33% of the area of the box filled by cells) were discarded to eliminate noise along the tissue border. Third, each grid box was considered an 'ROI' and then tracked through the entire movie using TissueMiner's ROI tracking code. Fourth, the rate of deformation by each type of cellular contribution was calculated as a moving average (kernel = 11) for each grid position for each timestep. Last, we averaged these rates over all time points post-adaption period and plotted in space (*Figure 1H,I,K,L*).

### Plotting the radial profile

To show tissue deformation as a function of distance to the center, we first calculated the distance to the center of symmetry for each cell. Second, we divided the tissue visible in the last frame into radial bins, rather than a grid, by rounding the radial position of each cell to its nearest $10 \mu m$. Third, as we did for the grid, we defined each radial bin to be an 'ROI' and then tracked the region forward and backward using TissueMiner's ROI tracking code. Fourth, the rate of deformation by each type of cellular contribution was calculated as a moving average (kernel = 11) for each radial bin ROI at each timestep. Last, we averaged these rates over all time points post-adaption period and plotted this rate as a function of distance to the center. Note that we also show the spatial profiles of tissue growth in the band around the DV boundary in *Figure 1—figure supplement 1*, but here we binned along $x$, not $r$.

## Regional analysis of anisotropic tissue deformation

### Spatial maps of cellular contributions to anisotropic tissue deformation

We previously published patterns of cellular contributions to anisotropic tissue shear from movies 1 to 3 (*Dye et al., 2017*); however here, we calculated these patterns in a more accurate way and report averages over multiple movies (*Figure 2A*). Previously, we assigned a grid at each timepoint as in *Etournay et al., 2015*; *Etournay et al., 2016*. While this method provides a simple first approximation of the pattern, it is not completely accurate because we are not tracking the box in time but reassigning it at each timepoint; thus, cell movement in/out of the box is not counted. Here, we perform grid box tracking, as described above for the calculation of cellular contributions to isotropic tissue deformation but with grid box size = $15 \mu m$. Further, we present a global average of not just movies 1 to 3, but also the two new movies. We calculated for each movie the rate of tissue deformation by each cellular contribution as a moving average (kernel = 11) in time. Then, we averaged across all five movies at each timepoint and presented the pattern as a cumulative sum, starting from the end of the adaption phase (first $2hr$) and continuing through the end of the timelapse. All correlation terms were added together (see *Merkel et al., 2017*). The contribution to tissue shear from cell extrusion is very small and thus not shown.

### Accumulating cellular contributions to radial tissue shear

We calculated a moving average (kernel size = 11) total value for each cellular contribution to radial tissue shear, averaged across the blade (excluding the band around the DV boundary). Then, we accumulated the contributions after the adaption phase (first $2hr$) and plotted over time (*Figure 2B*). In addition to the previously described types of correlations (*Merkel et al., 2017*), the radial decomposition also involves a correlation between cell area and shear (see Appendix 3). We added all correlation terms together in *Figure 2B*.

## Quantification of circular laser ablation

### Measurement of stress, cell elongation, and cell area

For the ablation data, we first projected the Z-stack images of the wing discs before and after ablation using our custom surface projection (*Dye et al., 2017*).

To quantify cell elongation and area, we used the images taken before the cut. Cells were segmented using the FIJI plugin TissueAnalyzer and then processed with TissueMiner to rotate the disc to a common orientation (anterior to the left; dorsal up) and to create a triangle network and a database structure. The area-weighted average of triangle elongation and the average cell area was calculated for those cells included in the center of a circle of a radius $r = 1.3 \times r_{cut}$ ($4.55 \mu m$) centered at the center of the cut. Varying the size of this region only slightly affects the result: $< 1.0 \times r_{cut}$, the distribution becomes more noisy because we are averaging a smaller region with less cells, but regions that are too big ($\sim 1.8 \times r_{cut}$) may begin to include cells that span different regions of the wing (i.e. the band around the DV boundary). In *Figure 4C* and *Figure 4—figure supplement 1B,E*, we plot cell elongation projected onto the stress axis.

To measure the shape of the tissue after ablation, we fit ellipses to the inner and outer piece left by the cut using the projected after-cut images. These images were first rotated using the transformation performed on the before-cut images to orient the tissue. We used Ilastik (v. 1.2.2) (*Berg et al., 2019*) to segment the cut region: we trained the classifier using all the data from a single day's acquisition, delineating three regions: cells, membranes, and dark regions. Using the trained classifier, we then generated a thresholded binary image of the cut tissue's shape and cropped the image around the cut. We then fit inner and outer ellipses to these images.

## Grouping ablations into regions of the wing pouch

To quantify the position of the cut in the wing pouch, the immunofluorescence post-cut images were projected using maximum intensity. In FIJI, we manually measured (in $\mu m$) the distance from the center of the cut to the DV boundary and AP boundary. We noted that fixation causes the tissue to shrink. We estimated the extent of shrinkage using a subset of the samples in which the compartment boundaries were visible in the Z-stack taken of the live sample before the cut. We used FIJI to measure distances from the center of the cut to the compartment boundaries in both the pre-fix live images and in the post-fix immunofluorescence images for this subset. We measure a discrepancy indicating that the tissue shrinks by ~15%. Thus, we compensate for this shrinkage when measuring distances to the compartment boundaries in our analysis. We also noted the compartment (dorsal/ventral/anterior/posterior) in post-fix images and added a sign to the distances to the boundaries (ventral and posterior getting negative 'distances') to create the spatial map shown in *Figure 4B*.

To classify the cuts by position, we first estimated the width of the band around the DV boundary from the timelapse data to be $\sim 22 \mu m$ (centered on the DV boundary). We then classified a cut as in the DV boundary if its cut boundary extended no more than $1.4 \mu m$ (20% of the $7 \mu m$ radius) outside this $22 \mu m$ horizontal stripe region. Likewise, cuts were classified as outside this region if it did not extend more than $1.4 \mu m$ into the $22 \mu m$ horizontal stripe. We excluded all cuts that appear to straddle the border between these regions (gray in *Figure 4B*). We also excluded those cuts with centers lying within $7 \mu m$ of the AP boundary, in case material properties at this boundary region are also different.

## ESCA: determination of stress from response to circular ablation

To obtain the normalized shear stress $\tilde{\sigma}/(2K)$, normalized isotropic stress $\sigma/\overline{K}$ and the ratio of elastic constant $2K/\overline{K}$, we fit ellipses to the inner and outer tissue outlines simultaneously. These three parameters determine the large and small semi-axes of the two ellipses. Other fitting parameters are the center positions of the two ellipses and the angles of major axes. Stress is calculated by considering how the ellipses were generated by a circular laser ablation, as described in detail in the Appendix 2. When calculating the deformation from initial to final positions, we considered that the cells that were directly hit by the laser would die and thus not contribute to the evolution of the tissue after the cut. To account for this loss of cells, we first calculated average cell area for those cells that would be hit by the laser. Then, we adjusted the radius of the initial position by this amount.

Fits were performed on all of the laser ablations in the same region (either within or outside of the band around the DV boundary) for a range of fixed values of the ratio $2K/\overline{K}$. The optimal value was considered to be the one that minimizes the sum of fit residuals (*Figure 4—figure supplement 1C,F*). We defined a threshold of fit residual, as shown in *Figure 4—figure supplement 1A*, to eliminate cuts with non-elliptical outlines. The uncertainty of $2K/\overline{K}$ was estimated by bootstraping a subset of 7 cuts in each region 100 times; we report the standard deviation of the results.

## Acknowledgements

We thank Stephan Grill for the microscope used to perform circular laser ablations. We thank the Light Microscopy Facility of the MPI-CBG for assistance with all other imaging experiments. Benoit Lombardot, formerly of the Image Analysis Facility of the MPI-CBG, provided us with a FIJI script to perform the projection of the apical surface, originally designed by Dagmar Kainmueller. Franz Gruber generated the *pk30,ecadGFP* fly line. We thank Christian Dahmann, Charlie Duclut, Kinneret Keren, Yonit Maroudas-Sacks, Alphee Michelot, and Ioannis Nellas for critical review of the manuscript. Lastly, we dedicate this manuscript to our co-author, Suzanne Eaton, who passed away tragically toward the conclusion of this work. This work was supported by funding from the Max-Planck-Gesellschaft (NAD, KVI, JFF, RPG, SE, FJ), Deutsche Forschungsgemeinschaft (EA4/10-1, EA4/10-2; NAD, KVI, SE), Swiss National Science Foundation (Grant #200021–165509; MP), the Simons Foundation (Grant #454953 to Matthieu Wyart; MP), the Deutsche Krebshilfe/MSNZ Dresden (NAD), and the ELBE PhD program (RPG). Funding sources were not involved in the study design, data collection, data interpretation, or the decision to submit the work for publication.

## Additional information

### Funding

| Funder | Grant reference number | Author |
| --- | --- | --- |
| Max-Planck-Gesellschaft | | Natalie A Dye<br>Marko Popovic<br>K Venkatesan Iyer<br>Jana Fuhrmann<br>Romina Piscitello-Gómez<br>Suzanne Eaton<br>Frank Jülicher |
| Deutsche Forschungsgemeinschaft | EA4/10-1 | Natalie A Dye<br>K Venkatesan Iyer<br>Suzanne Eaton |
| Swiss National Science Foundation | 200021-165509 | Marko Popovic |
| Simons Foundation | 454953 | Marko Popovic |
| Deutsche Forschungsgemeinschaft | EA4/10-2 | Natalie A Dye<br>K Venkatesan Iyer<br>Suzanne Eaton |
| ELBE PhD program | | Romina Piscitello-Gómez |
| Deutsche Krebshilfe | MSNZ Dresden | Natalie A Dye |

The funders had no role in study design, data collection and interpretation, or the decision to submit the work for publication.

### Author contributions

Natalie A Dye, Conceptualization, Resources, Data curation, Software, Validation, Investigation, Visualization, Methodology, Writing - original draft, Project administration, Writing - review and editing; Marko Popović, Conceptualization, Data curation, Software, Formal analysis, Funding acquisition, Validation, Visualization, Methodology, Writing - original draft, Project administration, Writing - review and editing; K Venkatesan Iyer, Investigation, Methodology, Writing - review and editing; Jana F Fuhrmann, Investigation, Writing - review and editing; Romina Piscitello-Gómez, Validation, Investigation, Writing - review and editing; Suzanne Eaton, Conceptualization, Resources, Supervision, Funding acquisition, Methodology, Project administration; Frank Jülicher, Conceptualization, Resources, Supervision, Funding acquisition, Methodology, Writing - original draft, Project administration, Writing - review and editing

## Author ORCIDs

Natalie A Dye (iD) https://orcid.org/0000-0002-4859-6670
Marko Popović (iD) https://orcid.org/0000-0003-2360-3982
Frank Jülicher (iD) https://orcid.org/0000-0003-4731-9185

## Decision letter and Author response

Decision letter https://doi.org/10.7554/eLife.57964.sa1
Author response https://doi.org/10.7554/eLife.57964.sa2

## Additional files

### Supplementary files

- Supplementary file 1. Description of data tables provided in Source Data.

- Transparent reporting form

### Data availability

We have made all data analyzed during this study available. Data for Figures 1H–M, 2,4,5, and 7 are provided as source data files. The data on cell area and elongation in Figure 1A–F, 3F,G, and 6C are too large to be submitted here and are available on Dryad (https://doi.org/10.5061/dryad.jsxksn06b).

The following dataset was generated:

| Author(s) | Year | Dataset title | Dataset URL | Database and Identifier |
|---|---|---|---|---|
| Dye NA | 2021 | Data from: Self-organized patterning of cell morphology via mechanosensitive feedback | http://dx.doi.org/10.5061/dryad.jsxksn06b | Dryad Digital Repository, 10.5061/dryad.jsxksn06b |

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

## Appendix 1

### Continuum model of the wing disc epithelium

### 1 Model definition

We use a previously developed continuum model to describe tissue mechanics (*Etournay et al., 2015*; *Popović et al., 2017*). Tissue shear flow $\tilde{v}_{ij}$ consists of convected co-rotational change of cell elongation $Q_{ij}$ and a contribution from cell rearrangements $R_{ij}$

$$\tilde{v}_{ij} = \frac{DQ_{ij}}{Dt} + R_{ij} \quad . \tag{A1}$$

Tissue shear stress consists of an elastic part and a contribution from nematic cell polarity $q_{ij}$

$$\tilde{\sigma}_{ij} = 2KQ_{ij} + \zeta q_{ij} \quad . \tag{A2}$$

Shear flow due to cell rearrangements consists of term accounting for cell shape relaxation and a contribution due to nematic cell polarity $q_{ij}$

$$R_{ij} = \frac{1}{\tau} Q_{ij} + \lambda q_{ij} \quad . \tag{A3}$$

Force balance couples shear stress and area pressure

$$\partial_j \tilde{\sigma}_{ij} - \partial_i P = 0 \tag{A4}$$

which will be reflected in cell area. We use a constitutive relation

$$P = -\overline{K} ln\left(\frac{a}{a_0}\right) \tag{A5}$$

where $a$ represents cell area and $a_0$ corresponds to value of $a$ at $P = 0$.

We consider a simple model in which the tissue is a radially symmetric disc, see *Figure 3C–E* of the main text. We use polar coordinate system $(r, \varphi)$, where the radial force balance reads

$$\partial_r P = \partial_r \tilde{\sigma}_{rr} + \frac{2}{r} \tilde{\sigma}_{rr} \tag{A6}$$

Furthermore, in the model we consider a constrained tissue with negligible radial shear flow $\tilde{v}_{rr} = 0$, consistent with experimental observations, see *Figure 2* of the main text.

### 2 Cell area profiles follow from cell elongation profiles

Empirically, the radial cell elongation profile can be described by a power law

$$Q_{rr} \simeq -\left(\frac{r}{r_Q}\right)^{\theta} \quad , \tag{A7}$$

with values for $r_Q = 140 \pm 20 \mu m$ and $\theta = 1.6 \pm 0.1$ obtained by fitting the data on the last $5hr$ of the experiments, see red line in *Appendix 1—figure 1A*. We find $r_Q = 84 \pm 3 \mu m$ and $\theta = 2.8 \pm 0.3$ by fitting the data from the first $\sim 5hr$. Data is averaged over five experiments, and errorbars represent a standard deviation among the mean values in each experiments.

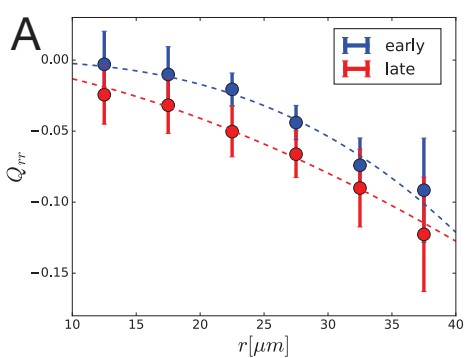
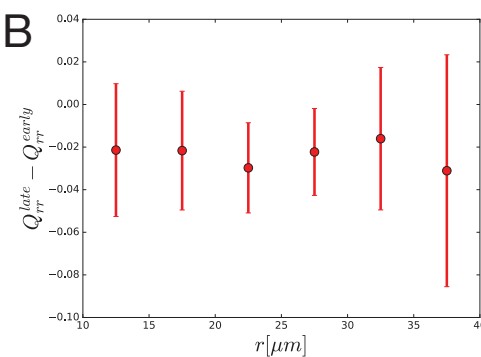

**Appendix 1—figure 1.** Dynamics of radial cell elongation profile. Left: Radial cell elongation in the first half of the movies ('early') are shown in blue and in the second half of the movies ('late') are shown in red. Fits of power law are shown with dashed lines in corresponding colors, with fit parameters $r_Q = 84 \pm 3\mu m$, $\theta = 2.8 \pm 0.3$ and $r_Q = 140 \pm 20\mu m$, $\theta = 1.6 \pm 0.1$ for "early" and "late" profiles, respectively. Right: Radial cell elongation change suggests that there is no significant gradient of radial shear due to cell rearrangements $R_{rr}$ during the experiment.

At this point, it is useful to estimate the spatial profile of shear due to cell rearrangements $R_{rr}$. We calculate the difference between late and early cell elongation profiles, as shown in **Appendix 1—figure 1B**. Since $\tilde{v}_{rr} \approx 2 \cdot 10^{-3} h^{-1}$ we find $R_{rr} \approx (6 \pm 7) \cdot 10^{-3} h^{-1}$ with no significant variation in $r$.

Now, we find an expression for cell area using **Equations A2, A3, A5, A6 and A7**

$$a = a_0 e^{-\frac{c_a}{\overline{K}} + \frac{2K^*\theta + 2}{\overline{K}}\frac{1}{\theta}\left(\frac{r}{r_Q}\right)^\theta - \frac{2K}{\overline{K}}\left(1 - \frac{K^*}{K}\right)\tau R_{rr} \ln\left(\frac{r}{r_Q}\right)} \,, \tag{A8}$$

where

$$K^* = \left(1 - \frac{\zeta}{2K\tau\lambda}\right)K \tag{A9}$$

and $c_a$ is an integration constant. We fit the **Equation A8** to the experimental data as shown in **Figure 3G** of the main text, and we find parameter values

$$a_0 e^{-\frac{c_a}{\overline{K}}} = 3.3 \pm 0.3 \quad , \tag{A10}$$

$$\frac{2K^*}{\overline{K}}\frac{\theta + 2}{\theta} = 5.4 \pm 0.7 \quad , \tag{A11}$$

$$\frac{2K}{\overline{K}}\left(1 - \frac{K^*}{K}\right)\tau R_{rr} = 0.00 \pm 0.07 \quad , \tag{A12}$$

where the reported uncertainties represent square roots of the fit covariance matrix and the parameter in **Equation A12** was constrained to be positive, motivated by the fact that we do not find $K^* > K$ (see **Equation A18** below).

## 3 Laser ablation experiments provide an estimate of model parameters

We write **Equation A2** in the form

$$\frac{\tilde{\sigma}_{rr}}{2K} = \frac{K^*}{K}Q_{rr} + \left(1 - \frac{K^*}{K}\right)\tau R_{rr} \quad . \tag{A13}$$

We fit this equation to the experimentally obtained shear stress as a function of cell elongation in the laser ablation experiments. We find

$$\frac{K^*}{K} = 0.05 \pm 0.02 \quad , \tag{A14}$$

$$\left(1 - \frac{K^*}{K}\right)\tau R_{rr} = 0.011 \pm 0.002 \quad . \tag{A15}$$

Using the estimate for $R_{rr}$ we find

$$\tau = (2 \pm 2)hr \quad . \tag{A16}$$

This value is consistent with the one found for the cell elongation relaxation time-scale in the pupal wing (*Etournay et al., 2015*).

In *Equation A3,* we can now show that cell elongation profile in the wing pouch reflects the profile of nematic cell polarity. Namely, $|Q_{rr}| \gg \tau|R_{rr}| \approx 0.012$ in most of the tissue, except in the vicinity of the DV boundary, and thus

$$q_{rr} \approx -\tau\lambda Q_{rr} \quad . \tag{A17}$$

## Comparison of laser ablation and area profile fits

It is interesting to notice that from *Equation A11* and the value for $2K/\overline{K} = 3.4 \pm 0.4$ obtained from the laser ablation experiments (main text and Materials and methods), we can estimate

$$\frac{K^*}{K} = 0.6 \pm 0.1 \quad . \tag{A18}$$

This value is significantly higher than the one obtained by a direct fit to the laser ablation data (*Equation A14*). This apparent discrepancy stems in part from the fact that the elastic constant $\overline{K}$ in *Equation A5* is a non-linear elastic modulus. It can only be related to the one used in the laser ablation method by linearizing it around a typical value of $a$ in the data. When this value is not equal to $a_0$ the linearized elastic constant will differ from the original $\overline{K}$ in *Equation A5*. Furthermore, *Equation A5* does not take into account any adaptation of cell area to pressure or nematic cell polarity, which could influence the measured value $K^*/K$ when fitting the cell area profile. However interesting, these effects are beyond the scope of our current work.

## 4 Mechanosensitivity leads to spontaneous cell polarity by self-organization
### a Homogeneous tissue

We propose a dynamical equation of the polarity tensor $q_{ij}$

$$\frac{Dq_{ij}}{Dt} = -\frac{1}{\tau_q}q_{ij} - \mu\tilde{\sigma}_{ij} \quad , \tag{A19}$$

where the parameter $\mu$ represents the mechanosensitive response of nematic cell polarity to shear stress. We use *Equations A1, A2, A3 and A19* to obtain a dynamic equation for shear stress

$$\frac{\partial\tilde{\sigma}_{ij}}{\partial t} = 2K\tilde{v}_{ij} - \left(\frac{1}{\tau} + \zeta\mu\right)\tilde{\sigma}_{ij} - \left(2K\lambda + \zeta\left(\frac{1}{\tau_q} - \frac{1}{\tau}\right)\right)q_{ij} \quad . \tag{A20}$$

The system of linear *Equations A19 and A20* becomes unstable when

$$2K^*\tau_q\mu > 1 \quad . \tag{A21}$$

This instability will result in a finite value of nematic cell polarity in the tissue. To account for this polarized state we have to include a stabilising higher order term in *Equation A19*

$$\frac{Dq_{ij}}{Dt} = -\frac{1}{\tau_q}q_{ij} - \mu\tilde{\sigma}_{ij} - \alpha q^2 q_{ij} \quad . \tag{A22}$$

In the steady state, the magnitude squared of the nematic cell polarity is

$$q^2 = q_0^2 \equiv \frac{2K^*\tau_q\mu - 1}{\alpha\tau_q} \quad . \tag{A23}$$

Here, we have introduced the strength of polarity $q_0$, corresponding to the magnitude of polarity in a homogeneous system. Note that the definition of $q_0^2$ allows for negative values, which would correspond to no spontaneous polarization in the tissue.

## b Polarization of a radially symmetric tissue

To account for spatial variations of nematic cell polarity, we introduce a Laplacian term in *Equation A22*

$$\frac{Dq_{ij}}{Dt} = -\frac{1}{\tau_q}q_{ij} - \mu\tilde{\sigma}_{ij} - \alpha q^2 q_{ij} + D\nabla^2 q_{ij} \tag{A24}$$

To fit the radial profile of $Q_{rr}$ observed in time-lapse experiments we account for non-vanishing $R_{rr}$ as well. Consistent with the observation that during the experiments $R_{rr}$ does not vary significantly in time (*Figure 2B* main text) we use approximation $\tau\partial_t R_{rr} \ll R_{rr}$. Then $q_{rr}$ satisfies

$$\partial_r^2 q_{rr} + \frac{1}{r}\partial_r q_{rr} - \frac{q_{rr}}{r^2} = \frac{\alpha}{D}\left(q_{rr}^3 - q_0^2 q_{rr}\right) + \frac{1}{D}\left(\frac{1}{\tau\lambda} + 2K\mu\tau\right)R_{rr} \quad, \tag{A25}$$

where the left hand side is the radial component of the Laplacian of $q_{ij}$ in the polar coordinate system. From the solution of this equation, the $Q_{rr}$ profile is obtained as $Q_{rr} \approx -\tau\lambda q_{rr}$. This quantity is fitted to the experimentally measured tangential cell elongation profile by optimizing four tissue parameters, as well as two boundary conditions $q_{rr}(r_{in})$ and $\partial_r q_{rr}(r_{in})$ required to solve *Equation A25*. Here $r_{in} = 10\mu m$ is the smallest radius at which we measure the tangential cell elongation profile in *Figure 6C* of the main text. Parameters obtained by the fit are reported in *Table 1* of the main text, and the fitted $Q_{rr}$ profile is shown in *Figure 6* of the main text. Parameter uncertainties were estimated as the 10[th] and 90[th] percentile intervals obtained by fitting uniformly sampled profiles of $Q_{rr}$ from the interval $Q_{rr}(r) + \Delta Q_{rr}(r)$ and $Q_{rr}(r) - \Delta Q_{rr}(r)$, see *Figure 3* of the main text, with 101 realisations. Obtained uncertainty intervals are reported in the *Table 1* of the main text.

## c Predictions for experiments

Spontaneous cell polarity in our model depends on mechanosensitive feedback characterized by the feedback strength $\mu$. The characteristic polarity strength $q_0$ grows monotonically with $\mu$. A prediction from our model is therefore that when $\mu$ is reduced, cell elongation will be reduced because it is mediated by cell polarity. In order to test this prediction, we performed MyoVI knockdown experiments, which indeed show a decrease in cell elongation (*Figure 7*).

Our qualitative prediction is rather robust and does not depend on details of the system, such as the influence of compartment boundaries that disrupt the radial symmetry. However, boundary conditions, which we do not know, could have a significant influence on the system. Therefore, a full quantitative prediction of experiments is difficult. To show the self-consistency of our prediction, we test if the value of $q_0$ that we use to describe the data is consistent with the actual cell elongations observed. Indeed we have $q_0 \approx 0.1$, see *Table 1* which predicts a typical strength of cell elongation $Q_0 = -\lambda\tau q_0 \approx -0.1$, consistent with experimentally measured values, see *Figure 6C* of the main text. This suggests that our qualitative prediction is robust given the uncertainties in the system.

## Appendix 2

### Theory of circular laser ablations

We developed a method to infer the stresses in an epithelial tissue from the tissue boundary shape after a circular ablation. Here, we first briefly review linear elasticity in polar coordinates. Then, we derive equations relating the shape of the inner and outer boundaries obtained by the ablation of an infinite elastic sheet in the small deformation regime. The solution to this problem is by no means new, see for example (*Muskhelishvili, 1977*). Here, we present our derivation of the solution. The application of the method and the obtained parameter estimates are described in the Materials and methods section.

### 1 Linear elasticity in polar coordinates

#### a Displacement gradient

The deformation of an elastic object is described by the displacement vector of each point on the sheet

$$u_i(x_j) = x_i'(x_j) - x_i \tag{B1}$$

where $x_i$ is the intial position of a point on the elastic sheet and $x_i'$ is the position of that point after the deformation. Shape changes are described by the symmetric traceless gradient of displacement

$$u_{ij} = \frac{1}{2}\left(\partial_j u_i + \partial_i u_j\right) \quad, \tag{B2}$$

which can be decomposed into the isotropic part $u = u_{kk}$ and the shear part $\tilde{u}_{ij} = u_{ij} - \delta_{ij}u/2$. Note that we use summation convention over repeated indices. In polar coordinates displacement gradient is given by

$$u_{rr} = \partial_r u_r \tag{B3}$$

$$u_{\varphi\varphi} = \frac{1}{r}\partial_\varphi u_\varphi + \frac{1}{r}u_r \tag{B4}$$

$$u_{r\varphi} = \frac{1}{2}\left(\partial_r u_\varphi - \frac{1}{r}u_\varphi + \frac{1}{r}\partial_\varphi u_r\right) \tag{B5}$$

#### b Linear elastic constitutive relation

We are considering the linear elastic constitutive relation

$$\sigma_{ij} = 2K\tilde{u}_{ij} + \zeta q_{ij} + \overline{K}u\delta_{ij} \quad, \tag{B6}$$

where $\tilde{\sigma}_{ij}$ is the traceless symmetric component of shear stress, with shear elastic modulus $K$ and bulk elastic modulus $\overline{K}$. Here, we allow for the presence of active stresses due to a nematic field $q_{ij}$, corresponding to the nematic cell polarity in the wing disc. However, a constant active stress does not influence our results in the small deformation limit, since it simply redefines the reference configuration. We can therefore write this constitutive equation as

$$u_{ij} = \frac{1}{2K}\tilde{\sigma}_{ij} + \frac{1}{4\overline{K}}\sigma_{kk}\delta_{ij} \quad. \tag{B7}$$

#### c Force balance and compatibilty: Airy stress function

Force balance imposes

$$\partial_j\sigma_{ij} = 0 \quad. \tag{B8}$$

In two dimensions, force balance is satisfied by all stress fields whose components are derived from the Airy stress function $\Phi$

$$\begin{aligned}
\sigma_{xx} &= \partial_y^2 \Phi \quad, \\
\sigma_{xy} &= -\partial_x \partial_y \Phi \quad, \\
\sigma_{yy} &= \partial_x^2 \Phi \quad,
\end{aligned} \tag{B9}$$

which can be expressed in polar coordinates

$$\sigma_{rr} = \frac{1}{r}\partial_r \Phi + \frac{1}{r^2}\partial_\varphi^2 \Phi \quad, \tag{B10}$$

$$\sigma_{r\varphi} = -\partial_r\left(\frac{1}{r}\partial_\varphi \Phi\right) \quad, \tag{B11}$$

$$\sigma_{\varphi\varphi} = \partial_r^2 \Phi \quad. \tag{B12}$$

Now, a compatibility condition, which follows from a requirement of continuity of the displacement field, can be written as

$$\nabla^4 \Phi = 0 \quad. \tag{B13}$$

The general solution of this equation in polar coordinates, also called the Michell solution, is given by

$$\begin{aligned}
\Phi &= \left[A_0 r^2 + B_0 r^2 \ln r + C_0 \ln r\right] \\
&+ \left[I_0 r^2 + I_1 r^2 \ln r + I_2 \ln r + I_3\right]\varphi \\
&+ \left[A_1 r + B_1 r^{-1} + \overline{B}_1 r\varphi + C_1 r^3 + D_1 r \ln r\right]\cos\varphi \\
&+ \left[E_1 r + F_1 r^{-1} + \overline{F}_1 r\varphi + G_1 r^3 + H_1 r \ln r\right]\sin\varphi \\
&+ \sum_{n>1}\left[A_n r^n + B_n r^{-n} + C_n r^{n+2} + D_n r^{-n+2}\right]\cos n\varphi \\
&+ \sum_{n>1}\left[E_n r^n + F_n r^{-n} + G_n r^{n+2} + H_n r^{-n+2}\right]\sin n\varphi \quad.
\end{aligned} \tag{B14}$$

Inferring the stresses from laser ablation experiments consists of solving two elastic problems using the Michell solution as we now show.

## 2 Circular ablation of an elastic sheet under stress

Laser ablation splits the elastic sheet into inner and outer pieces. We now show that the boundary shapes of the two pieces are ellipses by solving the corresponding elastic problems.

### a Inner piece

We consider a piece of elastic sheet stretched by stress $\sigma_{xx} = \sigma_{xx}^0$, $\sigma_{yy} = \sigma_{yy}^0$, $\sigma_{xy} = 0$. After a circular laser ablation, the inner piece is under no stress. To infer the original stresses in the sheet, we determine the shape of the sheet boundary assuming the stresses are known. Then, given the boundary shape, we will be able to infer the stresses.

Stress in a sheet that is in mechanical equilibrium, under no external body force, is uniform throughout the sheet. Therefore, using *Equation B3* we can integrate *Equation B7* to find the components of the displacement vector $u_i$

$$u_r = \frac{r}{2K}\left[\tilde{\sigma}\cos 2\varphi + \frac{K}{2K}\sigma\right] \tag{B15}$$

$$u_\varphi = -\frac{r}{2K}\tilde{\sigma}\sin 2\varphi \quad, \tag{B16}$$

where we have introduced

$$\sigma = \sigma_{xx}^0 + \sigma_{yy}^0 \tag{B17}$$

$$\tilde{\sigma} = (\sigma_{xx}^0 - \sigma_{yy}^0)/2 \quad. \tag{B18}$$

To relate the displacement to the shape of the inner piece boundary, we have to solve *Equation B1* for $r(\varphi)$, in the configuration illustrated in *Appendix 2—figure 1*

$$r\hat{r} + u_r\hat{r} + u_\varphi\hat{\varphi} = R\hat{r}' \quad . \tag{B19}$$

Using the fact that

$$\hat{r}' \cdot \hat{r} = \cos(\varphi' - \varphi) \tag{B20}$$
$$\hat{r}' \cdot \hat{\varphi} = \sin(\varphi' - \varphi) \tag{B21}$$

we obtain

$$r\left(\varphi \Big| \frac{\sigma}{K}, \frac{\tilde{\sigma}}{K}\right) = \frac{R}{\sqrt{\left[1 + \frac{\sigma}{4K} + \frac{\tilde{\sigma}}{2K}\cos(2\varphi)\right]^2 + \left[\frac{\tilde{\sigma}}{2K}\sin(2\varphi)\right]^2}} \quad . \tag{B22}$$

To simplify the notation, we introduce

$$\tilde{s} = \frac{\tilde{\sigma}}{2K} \tag{B23}$$
$$s = \frac{\sigma}{K} \quad , \tag{B24}$$

and we write the solution in the form

$$r(\varphi | s, \tilde{s}) = \frac{R}{\sqrt{\left[1 + \frac{1}{4}s + \tilde{s}\cos(2\varphi)\right]^2 + \left[\tilde{s}\sin(2\varphi)\right]^2}} \quad . \tag{B25}$$

We denote semiaxes of this ellipse along the $x$ and $y$ axes, as defined in *Appendix 2—figure 1*, by $a_{\text{in}}$ and $b_{\text{in}}$, respectively. We find that

$$a_{\text{in}} = \frac{R}{\left|1 + \frac{1}{4}s + \tilde{s}\right|} \tag{B26}$$
$$b_{\text{in}} = \frac{R}{\left|1 + \frac{1}{4}s - \tilde{s}\right|} \tag{B27}$$

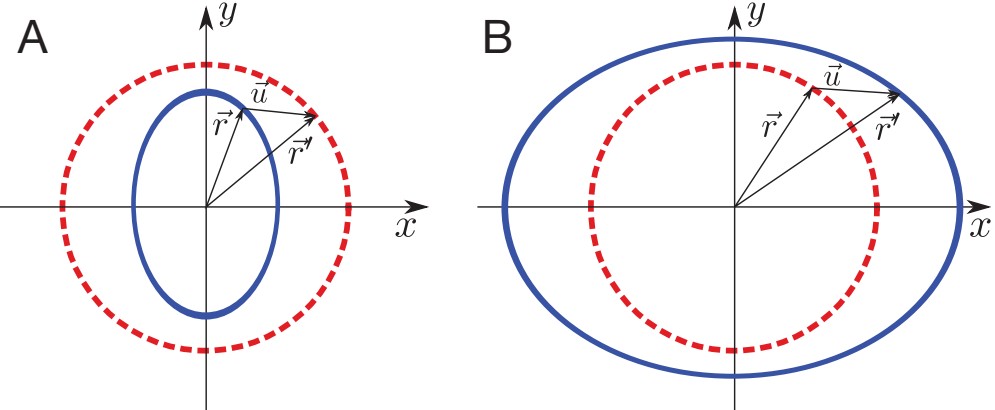

**Appendix 2—figure 1.** Schematic of elastic sheet boundary shape after a circular ablation. (**A**) Inner piece problem: Blue ellipse shape of the boundary after the ablation is the reference shape deformed by uniform stress $\sigma_{xx} = \sigma_{xx}^0$, $\sigma_{yy} = \sigma_{yy}^0$ to the red circle boundary shape before the ablation. (**B**) Outer piece problem: The red circle boundary shape before the ablation is the reference in this case. The final boundary shape after the ablation is blue ellipse.

## b Outer piece

As in the case of the inner piece, we need to find the boundary shape as a function of the original stresses in the sheet. In this case, the normal stress at the ablation boundary is zero, but the sheet is not stress-free. To simplify the problem, we consider an initially stress-free sheet with a circular hole to which boundary stresses $-\sigma_{ij}^0$ are applied. For small deformations, where the elastic problem is

linear and solutions of the problems commute, the boundary shape is equivalent to the one created by the laser ablation of a sheet under stress $\sigma_{ij}^0$.

Now, we have to solve the full elastic problem, since stresses are not uniform. We keep the terms of the Airy stress function $\Phi$, which obey nematic symmetry of the problem and relax sufficiently fast at infinity. We also assume that the polar axis is oriented along one of the principal axes of stress tensor, such that the off-diagonal component of stress vanishes. We obtain

$$\Phi = C_0 \ln r + \left( D_2 + \frac{B_2}{r^2} \right) \cos(2\varphi) \tag{B28}$$

and for stress components, using **Equation B10**

$$\sigma_{rr} = \frac{C_0}{r^2} - 2\left( 2\frac{D_2}{r^2} + 3\frac{B_2}{r^4} \right) \cos(2\varphi) \tag{B29}$$

$$\sigma_{r\varphi} = -2\left( \frac{D_2}{r^2} + 3\frac{B_2}{r^4} \right) \sin(2\varphi) \tag{B30}$$

$$\sigma_{\varphi\varphi} = -\frac{C_0}{r^2} + 6\frac{B_2}{r^4} \cos(2\varphi) \tag{B31}$$

The stress boundary condition reads

$$\sigma_{rr}(r=R) = -\frac{\sigma}{2} - \tilde{\sigma}\cos(2\varphi) \tag{B32}$$

$$\sigma_{r\varphi}(r=R) = \tilde{\sigma}\sin(2\varphi) \tag{B33}$$

allows us to determine coefficients in **Equation B29**

$$C_0 = -\frac{1}{2}\sigma R^2 \tag{B34}$$

$$B_2 = -\frac{1}{2}\tilde{\sigma}R^4 \tag{B35}$$

$$D_2 = \tilde{\sigma}R^2 \quad . \tag{B36}$$

Now we can calculate and then integrate the deformation gradient components to obtain

$$u_r = R\left[ \frac{\overline{K}}{4K}\frac{\sigma}{\overline{K}}\frac{R}{r} + \left[ 2\left(1+\frac{K}{\overline{K}}\right)\frac{R}{r} - \frac{R^3}{r^3} \right] \frac{\tilde{\sigma}}{2K}\cos(2\varphi) \right] \tag{B37}$$

$$u_\varphi = -R\left[ 2\frac{K}{\overline{K}}\frac{R}{r} + \frac{R^3}{r^3} \right] \frac{\tilde{\sigma}}{2K}\sin(2\varphi) \quad . \tag{B38}$$

Finally, as for the inner cut, we have to solve

$$R\hat{r} + u_r\hat{r} + u_\varphi\hat{\varphi} = r'\hat{r}' \tag{B39}$$

for $r'(\varphi')$. We express the solution using the angle $\varphi$ as a parameter

$$\varphi'\left( \varphi | \frac{\sigma}{\overline{K}}, \frac{\tilde{\sigma}}{2K}, \frac{K}{\overline{K}} \right) = \varphi - \arctan\left( \frac{\left(1+2\frac{K}{\overline{K}}\right)\frac{\tilde{\sigma}}{2K}\sin(2\varphi)}{1+\frac{\overline{K}}{4K}\frac{\sigma}{\overline{K}} + \left(1+2\frac{K}{\overline{K}}\right)\frac{\tilde{\sigma}}{2K}\cos(2\varphi)} \right) \tag{B40}$$

$$r'\left( \varphi | \frac{\sigma}{\overline{K}}, \frac{\tilde{\sigma}}{2K}, \frac{K}{\overline{K}} \right) = R\sqrt{\left[ 1+\frac{\overline{K}}{4K}\frac{\sigma}{\overline{K}} + \left(1+2\frac{K}{\overline{K}}\right)\frac{\tilde{\sigma}}{2K}\cos(2\varphi) \right]^2 + \left[ \left(1+2\frac{K}{\overline{K}}\right)\frac{\tilde{\sigma}}{2K}\sin(2\varphi) \right]^2} \quad . \tag{B41}$$

Note the ratio of elastic constants appears as a parameter of shape.

Now we show that the shape defined by **Equations B40 and B41** is an ellipse. To this end, we first define

$$A = \left(1 + \frac{2K}{\overline{K}}\right)\frac{\tilde{\sigma}}{2K} \tag{B42}$$

$$B = 1 + \frac{\overline{K}}{4K}\frac{\sigma}{\overline{\overline{K}}} \quad . \tag{B43}$$

This allows us to write *Equations B40 and B41* as

$$\varphi'(\varphi) = \varphi - \arctan\left(\frac{A\sin(2\varphi)}{B + A\cos(2\varphi)}\right) \tag{B44}$$

$$r'(\varphi) = R\sqrt{A^2 + B^2 + 2AB\cos 2\varphi} \quad . \tag{B45}$$

Now we show that $r'(\varphi')$ is an ellpise. An ellipse equation can be written as

$$r'(\varphi') = \frac{a_{\text{out}} b_{\text{out}}}{\sqrt{b_{\text{out}}^2 \cos^2\varphi' + a_{\text{out}}^2 \sin^2\varphi'}} \quad , \tag{B46}$$

where $a_{\text{out}}$ and $b_{\text{out}}$ are semiaxes of the ellipse oriented along $x$ and $y$ axes, respectively. Starting from *Equation B46* and expressing $\varphi'$ from *Equation B44* we will show that we can recover *Equation B45* for a particular choice of parameters $a$ and $b$.

We first calculate

$$\cos\varphi' = \frac{(A + B)\cos\varphi}{\sqrt{A^2 + B^2 + 2AB\cos 2\varphi}} \tag{B47}$$

$$\sin\varphi' = \frac{(B - A)\sin\varphi}{\sqrt{A^2 + B^2 + 2AB\cos 2\varphi}} \quad . \tag{B48}$$

Inserting these identities into *Equation B46* we find

$$r'(\varphi') = \frac{a_{\text{out}} b_{\text{out}}\sqrt{A^2 + B^2 + 2AB\cos 2\varphi}}{\sqrt{b_{\text{out}}^2(A + B)^2\cos^2\varphi + a_{\text{out}}^2(B - A)^2\sin^2\varphi}} \quad . \tag{B49}$$

Now, we only need to find $a_{\text{out}}$ and $b_{\text{out}}$ for which

$$\frac{a_{\text{out}}^2 b_{\text{out}}^2}{R^2} = b_{\text{out}}^2(A + B)^2\cos^2\varphi + a_{\text{out}}^2(B - A)^2\left(1 - \cos^2\varphi\right) \quad , \tag{B50}$$

for all $\varphi$. The solutions are

$$a_{\text{out}} = R|A + B| \tag{B51}$$

$$b_{\text{out}} = R|B - A| \quad , \tag{B52}$$

or in terms of the original parameters

$$a_{\text{out}} = R\left|1 + \frac{\overline{K}}{4K}\frac{\sigma}{\overline{\overline{K}}} + \left(1 + \frac{2K}{\overline{K}}\right)\frac{\tilde{\sigma}}{2K}\right| \tag{B53}$$

$$b_{\text{out}} = R\left|1 + \frac{\overline{K}}{4K}\frac{\sigma}{\overline{\overline{K}}} - \left(1 + \frac{2K}{\overline{K}}\right)\frac{\tilde{\sigma}}{2K}\right| \quad . \tag{B54}$$

If, in addition to *Equations B23 and B24* we define

$$\xi = \frac{2K}{\overline{K}} \quad , \tag{B55}$$

we can write

$$a_{\text{out}} = R\left|1 + \frac{1}{2\xi}s + (1+\xi)\tilde{s}\right| \tag{B56}$$

$$b_{\text{out}} = R\left|1 + \frac{1}{2\xi}s - (1+\xi)\tilde{s}\right| \tag{B57}$$

*Equations B26, B27, B56 and B57* impose four constraints on three parameters: $s$, $\tilde{s}$, and $\xi$. When determining these parameters from experimental data we are in general not able to satisfy all four equations simultaneously. Instead, we find the three parameters that best fit the segmented outlines of tissue boundaries, see Materials and methods.

## Appendix 3

### Correlation between cell elongation and cell movement contributes to the tissue radial shear flow

In this work, we use the previously developed method to define and calculate the cell elongation tensor and tissue shear due to T1 transitions, cell divisions, and extrusions (*Etournay et al., 2015*; *Merkel et al., 2017*). However, we apply the method in the polar coordinate system. To make it fully consistent, we have to introduce a new correlation term $S_{ij}$, which arises in the polar coordinate system when translation and elongation are correlated. To demonstrate the origin and derive an expression for $S_{ij}$, we consider a tissue that translates over the time interval $\Delta t$ but does not change otherwise. The tissue is triangulated by connecting the neighboring cell centers, see *Merkel et al., 2017*. Initially, the angle of a triangle $k$ with with respect to a center point is $\varphi$, blue triangle in *Appendix 3—figure 1*. During the translation by $\Delta \vec{r} = \vec{r}(t + \Delta t) - \vec{r}(t)$, the radial angle of the triangles changes by a small angle $\Delta \varphi^k$. Therefore, the components of the triangle elongation tensor $\tilde{q}_{ij}^t$ in the polar coordinate system change by

$$\Delta \tilde{q}_{ij}^t = -2\epsilon_{ik}\tilde{q}_{kj}^t\Delta\varphi \quad . \tag{C1}$$

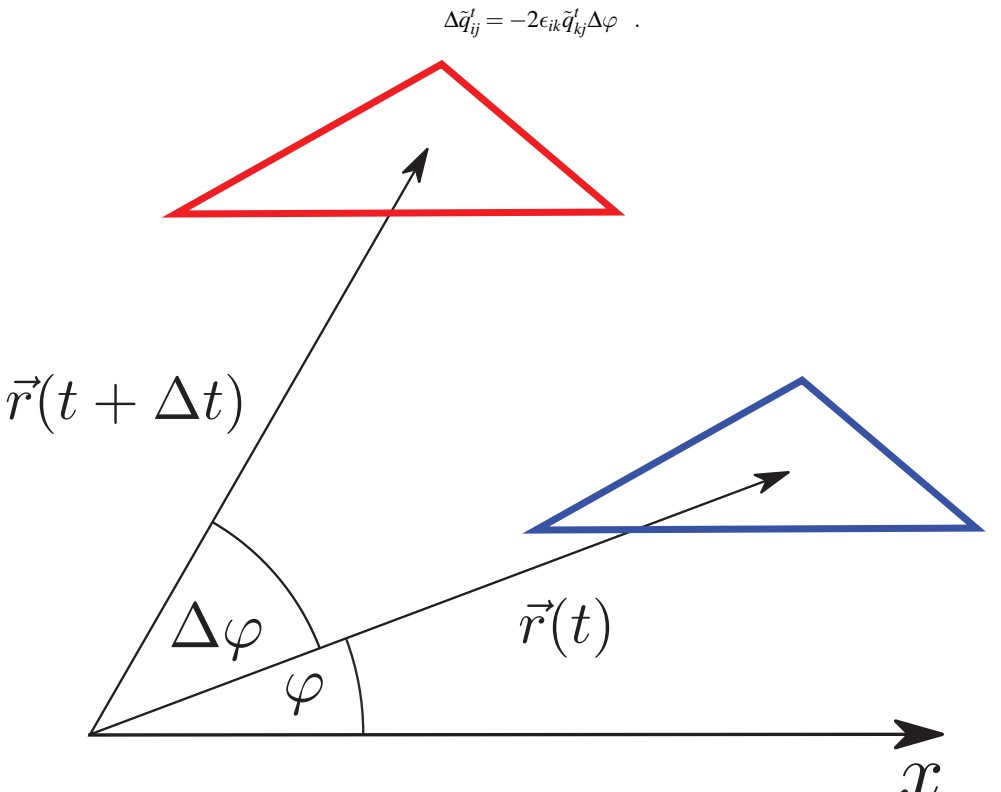

**Appendix 3—figure 1.** Triangle translation changes its radial elongation. Triangle is translated by $\Delta \vec{r} = \vec{r}(t + \Delta t) - \vec{r}(t)$. Although the triangle elongation did not change during the translation, the radial component of the triangle elongation changes due to the change of radial direction by $\Delta \varphi$.

However, since the triangles are only translating, there is no shear flow. Therefore, the average radial shear flow also vanishes and an additional term is necessary to account for the change in the average radial cell elongation in a translating tissue

$$\tilde{v}_{ij} = 0 = \frac{\Delta Q_{ij}}{\Delta t} + S_{ij} \quad . \tag{C2}$$

Therefore, the correlation term is given by

$$S_{ij} = \frac{1}{\Delta t}\left\langle 2\epsilon_{ik}\tilde{q}_{kj}^t\Delta\varphi \right\rangle \quad . \tag{C3}$$

In *Appendix 3—figure 2*, we show the new correlation term in black measured in the pouch of

the wing disc, averaged over five time-lapse experiments. For comparison, we show the correlation term stemming from correlated fluctuations of triangle rotation and elongation in blue and the correlation term stemming from correlated fluctuations of triangle area and elongation in green. For definitions and discussion of the other two correlation terms see *Etournay et al., 2015*; *Merkel et al., 2017*. We find that the contribution of the new correlation term to the radial shear flow is much smaller that the other two correlation terms. Therefore, the contribution to radial tissue shear flow of the correlation between translation and triangle elongation is negligible in the wing disc.

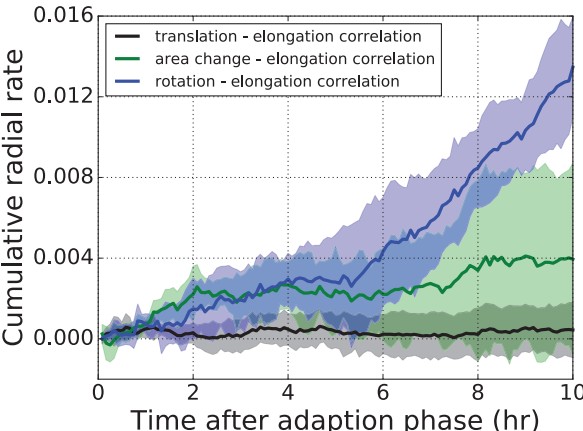

**Appendix 3—figure 2.** Three correlation terms contributing to the average radial tissue shear flow. The three correlation terms contributing to radial tissue shear flow measured in the wing disc pouch, averaged over five time-lapse movies.
Shaded regions correspond to one standard deviation of that sample. Blue: Correlation term stemming from correlated fluctuations of rotation and elongation of triangles. Green: Correlation term stemming from correlated fluctuations of area and elongation of triangles. Black: The new correlation term stemming from translation and elongation of triangles. The new correlation term (black) is very small in comparison to the other two.

