## [Decision Letter]

**Acceptance summary:**

This paper investigates the mechanisms governing the emergence of cell shape and size patterns in the *Drosophila* wing imaginal disc during larval development. They take advantage of their recently developed ex-vivo wing disc culture protocol and powerful image analysis tools, to obtain dynamic insight into a system that has traditionally been inaccessible to live imaging. Combining high-quality data, elaborate physical modeling and appropriate genetic perturbations, they challenge the current view that the patterns arise due to early differences in growth rate. In their alternative scenario, they propose that the mechanism driving cell morphology patterning is self-organized and originates in a mechano-sensitive feedback.

**Decision letter after peer review:**

Thank you for submitting your article "Self-organized patterning of cell morphology via mechanosensitive feedback" for consideration by *eLife*. Your article has been reviewed by 3 peer reviewers, one of whom is a member of our Board of Reviewing Editors, and the evaluation has been overseen by Aleksandra Walczak as the Senior Editor. The following individual involved in review of your submission has agreed to reveal their identity: Pierre Recho (Reviewer #2).

The reviewers have discussed the reviews with one another and the Reviewing Editor has drafted this decision to help you prepare a revised submission.

Summary:

In this study, Dye et al. investigate the mechanisms leading to the proximal-distal cell shape and size pattern in the wing disc pouch, in which the cells at the periphery of are bigger and more elongated than the cells at the center. They take advantage of their recently developed ex-vivo wing disc culture protocol (Dye et al., 2017) and powerful image analysis tools (Etournay et al., 2015, Merkel et al., 2017), to obtain dynamic insight into a system that has traditionally been inaccessible to live imaging. Using a combination of descriptive data, mathematical modeling and genetic perturbations, they challenge the current view that the pattern arises due to early differences in growth rate (Mao et al., 2013, LeGoff et al., 2013). In their alternative scenario, they propose that the mechanism driving the cell morphology patterning is self-organized and originates in a mechano-sensitive feedback. They base the theoretical framework on an elaborate phenomenological continuum model, which guides the different tests performed.

Essential revisions:

The quality of the descriptive data reported is remarkable and the phenomenological model is well grounded, appealing and to some extent predictive. The genetic tests performed provide significant evidence in support of the model but have some limitations. The emerging biophysical picture is novel, compelling and worthy of attention, as it implies a significant conceptual shift. Whenever fully validated, the mechanism here proposed will certainly bring fundamental biological insights into our understanding of cell morphology patterning. It has been a matter of debate among the reviewers, however, to what extent the direct experimental validation of some of the essential points of the study has been achieved, in order for the manuscript to be suitable for publication. The inherent limitations of phenomenological nature of the theoretical framework has also been raised as possibly reducing the breath of appeal, in the absence of a more direct and complete validation. From the reviewers' discussion, the following points have been raised which should be addressed in the revision to strengthen the manuscript.

1. On the polarity cue. The emergence of a polarity cue is pivotal in the proposed scenario, where it results from an instability induced by a mechano-sensitive feedback. This is a phenomenological hypothesis, which allows to explain the observations, and is to some extent predictive. However, the actual biochemical nature is not really identified, and it is not clear whether other scenarios could exhibit similar behavior. When the authors introduce cell polarity q, it would help to specify what physical quantity they have in mind, presumably associated to cytoskeletal orientation. In any case, it would be interesting that the authors suggest possible direct experimental measurements of such quantity, not relying on the model, for future validation.

2. On the role of mechano-sensitive coupling. A strong prediction of the model is that, for a finite value of mechano-sensitivity, the patterning should be abruptly suppressed, not just decreased. It would thus be very convincing to see that the patterning completely disappears by tuning the mechano-sensitive feedback beyond a critical (finite) value. Could the authors comment on why they think they cannot reach the complete disappearance of the patterning? Is it just an experimental limitation? Can other coexisting sources of polarization be taking over when the self-organized polarization is suppressed?

3. On the tissue constitutive equation. The authors mention in the introduction the fluidization mechanism described in Ranft et al., 2010, as a way to relax stresses in growing tissues. Could the authors explain why this mechanism is not present or is not relevant in the problem at hand? In connection with this point, the constitutive equation for the tissue is taken as purely elastic, while in general a growing tissue would generically be viscoelastic. Could the authors explain better the rationale for their choice?

4. On the Myosin VI regulation of the RhoA-ROCK-Myosin II pathway.

LeGoff et al. (2013) use the ROCK inhibitor Y27632 in the wing disc to deplete Myosin II from cell-cell junctions. Surprisingly, they show it does not affect the cell size or shape patterns in the wing pouch. We suggest the authors to comment on that, given they are proposing a mechanosensation mechanism based on ROCK-Myosin II regulation through Myosin VI.

5. PCP and polarization. Part of the authors' argument for the existence of a self-organised polarization process is that classical PCP is not involved. Their evidence is based on a negative result with perturbation of PCP by RNAi. RNAi is known to achieve incomplete gene activation and with a delay. Their approach should therefore be validated (e.g. by staining preparation with anti-Fmi). Alternatively, they could analyze previously validated mutant ds05142 or ft4/ftG-rv.

6. Myosin VI and mechanosensation. An important conclusion of this paper is that mechanosensation by Myosin VI is essential for polarization. Here the authors assume that Myosin IV is a mechano-sensor by extrapolating data from CaCo cells. There is no evidence that this is also true in their system. Of course, direct evidence (e.g. with a stretching device) that Myosin IV is part of the inferred mechanosensing machinery would be great, although we understand that this may take some effort. However, the authors could relatively easily verify that Myosin IV knockdown does affect Myosin II polarization, as predicted.

---

## [Author Response]

Essential revisions:The quality of the descriptive data reported is remarkable and the phenomenological model is well grounded, appealing and to some extent predictive. The genetic tests performed provide significant evidence in support of the model but have some limitations. The emerging biophysical picture is novel, compelling and worthy of attention, as it implies a significant conceptual shift. Whenever fully validated, the mechanism here proposed will certainly bring fundamental biological insights into our understanding of cell morphology patterning. It has been a matter of debate among the reviewers, however, to what extent the direct experimental validation of some of the essential points of the study has been achieved, in order for the manuscript to be suitable for publication. The inherent limitations of phenomenological nature of the theoretical framework has also been raised as possibly reducing the breath of appeal, in the absence of a more direct and complete validation. From the reviewers' discussion, the following points have been raised which should be addressed in the revision to strengthen the manuscript.1. On the polarity cue. The emergence of a polarity cue is pivotal in the proposed scenario, where it results from an instability induced by a mechano-sensitive feedback. This is a phenomenological hypothesis, which allows to explain the observations, and is to some extent predictive. However, the actual biochemical nature is not really identified, and it is not clear whether other scenarios could exhibit similar behavior. When the authors introduce cell polarity q, it would help to specify what physical quantity they have in mind, presumably associated to cytoskeletal orientation. In any case, it would be interesting that the authors suggest possible direct experimental measurements of such quantity, not relying on the model, for future validation.

The biochemical origin of the polarity cue that orients force-generating processes is indeed an important question. As part of an improved model presentation in the Results section, we now suggest actomyosin cytoskeletal polarization as a contribution to *ԛ*. MyoII, as well as PCP proteins, had already been shown experimentally to be planar polarized in the wing disc. We show in this work that the elimination of PCP does not result in a disrupted pattern of cell elongation. In contrast, we do think that MyoII, as a force-generating mechanosensitive motor, could be a significant component of *ԛ*, as suggested in the discussion of the original manuscript. We have now performed additional experiments (presented in Author response image 1) in response to point 6 supporting a contribution of MyoII to *ԛ*. However, the full molecular understanding of *ԛ* is beyond the scope of current manuscript. Thus, we prefer not to put these data in the manuscript.

**Author response image 1. sa2fig1:** MyoVI-RNAi disrupts polarity of MyoII. (**A**) Ecad-mCherry and fos-sqhGFP (MyoII regulatory light chain) expressing wings imaged under control and MyoVI-RNAi conditions. Note that the fosmid-based reporter of MyoII appears to be poorly expressed in a region just anterior to the anterior-posterior boundary in both conditions. (**B**) A closer view of the proximal anterior-dorsal region highlights the perturbation of anisotropy in the distribution of Ecad and MyoII. (**C**) We measure a full spatial pattern of MyoII-GFP nematic polarity in MyoVI-RNAi and control wings. We show an overlay of all measured patterns, revealing a less organized pattern in MyoVI-RNAi wings. (**D**) We further compare the average pattern of MyoII-GFP nematic cell polarity in the anterior region, marked by the yellow rectangle in (**C**), which is not affected by the poor expression of the MyoII reporter. We find that MyoVI-RNAi wings have an average pattern of MyoII polarity that appears to be reduced in magnitude and not entirely aligned to the control. To quantify these differences, we first compare in (**E**) the cumulative distribution of cell polarity magnitude C(q) in control wings with the component of polarity in MyoVI-RNAi wings along the axis defined locally by the average polarity in control wings. This quantification reveals a clear reduction of MyoII cell polarity in MyoVI-RNAi wings. Secondly, in (**F**) we show a comparison of the cumulative cell polarity magnitude in control and MyoVI-RNAi wings. The difference in magnitudes is smaller but significant, showing that the overall reduction of the MyoII pattern stems from both a reduction in magnitude and alignment relative to the control wings.

2. On the role of mechano-sensitive coupling. A strong prediction of the model is that, for a finite value of mechano-sensitivity, the patterning should be abruptly suppressed, not just decreased. It would thus be very convincing to see that the patterning completely disappears by tuning the mechano-sensitive feedback beyond a critical (finite) value. Could the authors comment on why they think they cannot reach the complete disappearance of the patterning? Is it just an experimental limitation? Can other coexisting sources of polarization be taking over when the self-organized polarization is suppressed?

We acknowledge that our original description was unclear on this point. In the spatially resolved model, which includes the polarity orientation coupling term (Figure 6C), a polarity pattern can exist even below 𝜇_𝑐_ if polarity exists at the tissue boundaries. Thus, the residual pattern of cell elongation that we observe after removal of MyoVI in the wing pouch could be due to polarity existing outside of this region. Furthermore, in our experiments, there are several additional considerations:

– Our knockdown of MyoVI by RNAi begins around 80hAEL and likely requires several hours to significantly reduce the levels of MyoVI. The ensuing relaxation of the pattern of cell shape will require additional time, and therefore it is possible that we are not observing the final state that results from complete knockdown. Our attempt to apply a stronger perturbation by co-expressing Dicer or using actin-gal4 (to express even earlier and stronger) resulted in severe growth defects and/or lethality (not shown).

– There may be additional sources of mechanosensitivity, such as vinculin. There may also, as the reviewer mentions, be other sources of cell polarity that can polarize the cytoskeleton and contribute to the patterning of cell shape in the absence of MyoVI. However, these additional factors are not sufficient to fully maintain the cell shape pattern upon inhibition of MyoVI, as we do detect a phenotype (shown in Figure 7).

– The tissue surrounding the wing disc pouch extends into the folds and is connected to the peripodial membrane. Therefore, we expect that boundary stresses and polarization of the boundary tissue could play an important role, as suggested above.

– The wing tissue is not exactly in a steady state during our timelapse experiments. To account for this fact, we used a constant rate of T1-induced shear as a fit parameter, see SI Equation 25, when fitting the model to the experimental data.

Thus, our inability to completely remove the cell shape pattern is largely an experimental limitation. We have now adjusted the corresponding Discussion section to clarify this point.

3. On the tissue constitutive equation. The authors mention in the introduction the fluidization mechanism described in Ranft et al., 2010, as a way to relax stresses in growing tissues. Could the authors explain why this mechanism is not present or is not relevant in the problem at hand? In connection with this point, the constitutive equation for the tissue is taken as purely elastic, while in general a growing tissue would generically be viscoelastic. Could the authors explain better the rationale for their choice?

In the continuum model (following (Etournay et al., 2015; Merkel et al., 2017)), we consider the wing tissue as an active, viscoelastic fluid and the cells as elastic objects. Elastic stresses are relaxed by topological changes via the term R, which include both cell rearrangements and rearrangements associated with cell divisions. The effect of cell divisions in our model corresponds to a contribution to the coefficient 1τ in Equation 3, of magnitude proportional to the cell division rate 𝑘_𝑑_. This rate is roughly constant in our experiments and therefore does not affect the form of Equation 3. Thus, our model is consistent with Ranft et al., and the fluidization effects by divisions are effectively contained in the parameter 1τ in Equation 3.

4. On the Myosin VI regulation of the RhoA-ROCK-Myosin II pathway.LeGoff et al. (2013) use the ROCK inhibitor Y27632 in the wing disc to deplete Myosin II from cell-cell junctions. Surprisingly, they show it does not affect the cell size or shape patterns in the wing pouch. We suggest the authors to comment on that, given they are proposing a mechanosensation mechanism based on ROCK-Myosin II regulation through Myosin VI.

We thank the reviewer for pointing out the relevance of this ROCK inhibitor experiment to our work. In (LeGoff et al., 2013), the authors indeed find that cell shape anisotropy in the periphery of the pouch is slightly but significantly *increased* after 40 minutes of exposure to Y27632 (Figure 5D in the reference). They state on p.4055 of this article that they, “do not favor a hypothesis linking MyoII cables and planar polarity pathways.” Instead, they “propose that MyoII polarity is a consequence of cell stretching” and “acts a negative mechanical feedback.” This conclusion is consistent with our observations and corresponds to our linearized mechanosensitive model. Our Equation 19 of SI describes the dependence of polarity on tissue shear stress (and consequently cell stretching). Equation 2 includes the “negative mechanical feedback,” whereby cell elongation is negatively influenced by changes in polarity on short timescales during which the stress does not relax by cell rearrangements. This equation is consistent with the stress changes observed after the 40min of Y27632 exposure. Namely, this 40 min incubation is likely shorter than the fluidization timescale 𝜏, which we estimate to be on the order of 2hr (see SI Equation 16). Interestingly, this experiment of LeGoff et al. provides additional information about the parameter values of our model, indicating that 𝜁 is positive in Equation 2 of SI. Our work goes beyond LeGoff et al. by suggesting that the cell morphology patterns emerge via a mechanosensitive instability in a self-organized process. We now comment on this point in the Discussion of the revised manuscript.

5. PCP and polarization. Part of the authors' argument for the existence of a self-organised polarization process is that classical PCP is not involved. Their evidence is based on a negative result with perturbation of PCP by RNAi. RNAi is known to achieve incomplete gene activation and with a delay. Their approach should therefore be validated (e.g. by staining preparation with anti-Fmi). Alternatively, they could analyze previously validated mutant ds05142 or ft4/ftG-rv.

We did not exclusively use RNAi to perturb PCP; we also used a previously characterized mutant of the core PCP component, Prickle. This mutation, *pk30*, eliminates the formation of the predominant Pk isoform that is expressed in the wing (Gubb et al., 1999). Our lab has previously shown that both the magnitude and orientation of Fmi polarity is reduced in *pk30* mutant cells in the wing (Merkel et al., 2014). Other labs have also shown that the removal of Prickle isoforms in the wing disrupts the planar polarized distribution of Dsh, Stbm, and Fz on the membrane (*pk-sple* mutant, Bastock et al., 2003; Strutt, 2001; Tree et al., 2002). To further verify that the core PCP pattern is not necessary to achieve the proper cell shape pattern in the wing disc, we also examined this pattern in a null mutant for a different core PCP component, Strabismus (*stbm8*). We still see a similar pattern of cell shapes (not shown).

To perturb the Fat/Ds pathway, we used RNAi lines that have been previously used in several other publications (Ambegaonkar and Irvine, 2015; Misra and Irvine, 2016, 2019; Rauskolb et al., 2011; Rogulja et al., 2008). The severe undergrowth that we observe upon simultaneous knockdown of both Fat and Dachs strongly suggests that this pathway is affected by the end of larval development, when we examined cell shapes (Figure 5). We have now also added data showing the growth defect in the adult wing (now in Figure 5—figure supplement 1). To further verify the effectiveness of the RNAi approach, we have also performed new experiments to stain for Dachsous upon simultaneous RNAi for Fat and Dachs. We find that Dachsous is largely lost from the apical membrane and any residual signal is not planar polarized. Thus, this RNAi approach is effective at removing this PCP pathway. We have added these data to a new supplemental figure (Figure 5—figure supplement 1). Note that the authors of the Legoff 2013 work discussed above also concluded that myosin polarity was not a consequence of Ft/Ds PCP (using a null mutation in Dachs), in agreement with our conclusions. The classical loss-of-function mutants in Fat/Ds suggested by the reviewer are severely overgrown and therefore are likely to have significantly different distribution of stresses, which would complicate the interpretation of any changes in cell shape pattern.

6. Myosin VI and mechanosensation. An important conclusion of this paper is that mechanosensation by Myosin VI is essential for polarization. Here the authors assume that Myosin IV is a mechano-sensor by extrapolating data from CaCo cells. There is no evidence that this is also true in their system. Of course, direct evidence (e.g. with a stretching device) that Myosin IV is part of the inferred mechanosensing machinery would be great, although we understand that this may take some effort. However, the authors could relatively easily verify that Myosin IV knockdown does affect Myosin II polarization, as predicted.

We performed new experiments, as suggested, to look for the effect of MyoVI knockdown on MyoII distribution. We show these supporting data in Author response image 1. We generated a fly line expressing Ecad-mCherry from its endogenous locus, as well as a GFP fusion to the regulatory light chain of MyoII from a fosmid (Sarov et al., 2016). Note that this fosmid-based reporter of MyoII appears to be poorly expressed in a region just anterior to the anterior-posterior boundary, even in control wing discs. Nonetheless, outside of this region, the signal resembles that reported for MyoII in the literature (LeGoff et al., 2013; Mao et al., 2013).

To quantify the polarity of MyoII, we define a structure tensor based on the gradients of MyoII-GFP signal intensity (De Vos et al., 2016), averaging in grids of 10𝑥10 𝜇𝑚, aligned on the AP and DV boundaries (see Method for Author response image 1 below). In Author response image 1, we show the superposition of the MyoII polarity profiles obtained in N=10 control experiments (top) and in N=9 MyoVI RNAi experiments (bottom). Upon MyoVI RNAi, the pattern of MyoII orientation is similar to control but appears noisier. We quantify this phenotype in the following panels. In D, we show the average MyoII polarity for each grid element, focusing on the anterior side of the wing to avoid the region of low MyoII signal. We find that MyoII polarity is reduced in the MyoVI RNAi wing discs for a majority of grid elements. We demonstrate that this difference is significant by comparing the cumulative distributions of MyoII polarity in MyoVI RNAi vs control in parts E-F. In E, the cumulative distribution of magnitudes of MyoII polarities in control discs is shown in blue. In yellow, we plot the cumulative distribution of MyoII polarities in RNAi discs, projected onto the axis defined by the MyoII polarity in control experiments in each grid element. The error bars show the standard deviation of the mean. We find that MyoII polarity is significantly perturbed, reflecting a reduced magnitude and an increased orientational disorder. Part F shows a comparison of the cumulative distributions of polarity magnitude.

This analysis clearly shows that MyoVI RNAi perturbs the polarity pattern of MyoII, consistent with the results shown in the main text. While these data are completely consistent with our suggestion that MyoII could significantly contribute to *ԛ*, after careful consideration we have decided to publish it only here in the response to reviewer comments. Precisely how MyoII affects the self-organization behavior deserves a more extensive investigation that is beyond the scope of this manuscript.

Method for Author response image 1:

Quantification of MyoII polarity using a structure tensor of GFP:

We divide the experimentally obtained images of MyoII-GFP signal into a rectangular grid starting at the intersection of AP and DV compartment boundaries with grid element size of 10𝑥10𝜇𝑚. In each of the grid elements we calculate a structure tensor *S* of the MyoII-GFP intensity 𝐼(𝑥, 𝑦) defined as:Sij=1A∫grid element∂iI(x,y)∂jI(x,y)dxdywhere A=∫grid elementdxdy is the area of a grid element and indices 𝑖 and 𝑗 correspond to 𝑥 and 𝑦.

Now, nematic polarity tensor *q* in each grid element is defined by:Sdet(S)=e2q

References:

Ambegaonkar, A. A., and Irvine, K. D. (2015). Coordination of planar cell polarity pathways through spiny-legs. *eLife*, *4*(OCTOBER2015). https://doi.org/10.7554/*eLife*.09946

Bastock, R., Strutt, H., and Strutt, D. (2003). Strabismus is asymmetrically localised and binds to Prickle and Dishevelled during *Drosophila* planar polarity patterning. *Development*, *130*(13), 3007– 3014. https://doi.org/10.1242/dev.00526

De Vos, W. H., Munck, S., and Timmermans, J.-P. (Eds.). (2016). *Focus on Bio-Image Informatics* (Vol. 219). Springer International Publishing. https://doi.org/10.1007/978-3-319-28549-8

Etournay, R., Popović, M., Merkel, M., Nandi, A., Blasse, C., Aigouy, B., Brandl, H., Myers, G., Salbreux, G., Jülicher, F., and Eaton, S. (2015). Interplay of cell dynamics and epithelial tension during morphogenesis of the *Drosophila* pupal wing. *eLife*, *4*, e07090.

https://doi.org/10.7554/*eLife*.07090

Gubb, D., Green, C., Huen, D., Coulson, D., Johnson, G., Tree, D., Collier, S., and Roote, J. (1999). The balance between isoforms of the prickle LIM domain protein is critical for planar polarity in *Drosophila* imaginal discs. *Genes and Development*, *13*(17), 2315–2327. https://doi.org/10.1101/gad.13.17.2315

LeGoff, L., Rouault, H., and Lecuit, T. (2013). A global pattern of mechanical stress polarizes cell divisions and cell shape in the growing *Drosophila* wing disc. *Development*, *140*(19), 4051–4059. https://doi.org/10.1242/dev.090878

Mao, Y., Tournier, A. L., Hoppe, A., Kester, L., Thompson, B. J., and Tapon, N. (2013). Differential proliferation rates generate patterns of mechanical tension that orient tissue growth. *The EMBO Journal*, *32*(21), 2790–2803. https://doi.org/10.1038/emboj.2013.197

Merkel, M., Etournay, R., Popović, M., Salbreux, G., Eaton, S., and Jülicher, F. (2017). Triangles bridge the scales: Quantifying cellular contributions to tissue deformation. *Physical Review E*, *95*(3), 032401. https://doi.org/10.1103/PhysRevE.95.032401

Merkel, M., Sagner, A., Gruber, F. S., Etournay, R., Blasse, C., Myers, E., Eaton, S., and Jülicher, F. (2014). The balance of prickle/spiny-legs isoforms controls the amount of coupling between core and fat PCP systems. *Current Biology*, *24*(18), 2111–2123.

https://doi.org/10.1016/j.cub.2014.08.005

Misra, J. R., and Irvine, K. D. (2016). Vamana Couples Fat Signaling to the Hippo Pathway. *Developmental Cell*, *39*(2), 254–266. https://doi.org/10.1016/j.devcel.2016.09.017

Misra, J. R., and Irvine, K. D. (2019). Early girl is a novel component of the Fat signaling pathway. *PLOS Genetics*, *15*(1), e1007955. https://doi.org/10.1371/journal.pgen.1007955

Rauskolb, C., Pan, G., Reddy, B. V. V. G., Oh, H., and Irvine, K. D. (2011). Zyxin links fat signaling to the hippo pathway. *PLoS Biology*, *9*(6), e1000624. https://doi.org/10.1371/journal.pbio.1000624

Rogulja, D., Rauskolb, C., and Irvine, K. D. (2008). Morphogen Control of Wing Growth through the Fat Signaling Pathway. *Developmental Cell*, *15*(2), 309–321. https://doi.org/10.1016/j.devcel.2008.06.003

Sarov, M. et al. (2016). A genome-wide resource for the analysis of protein localisation in *Drosophila*. *eLife*, *5*(FEBRUARY2016). https://doi.org/10.7554/*eLife*.12068

Strutt, D. I. (2001). Asymmetric localization of frizzled and the establishment of cell polarity in the *Drosophila* wing. *Molecular Cell*, *7*(2), 367–375. https://doi.org/10.1016/S1097-2765(01)001848

Tree, D. R. P., Shulman, J. M., Rousset, R., Scott, M. P., Gubb, D., and Axelrod, J. D. (2002). Prickle mediates feedback amplification to generate asymmetric planar cell polarity signaling. *Cell*, *109*(3), 371–381. https://doi.org/10.1016/S0092-8674(02)00715-8